# Threshold-adaptive pruning with multi-key homomorphic encryption for communication-efficient secure federated learning

Jie Guo[1], Renjing Liu[1*], Jinsheng Xing [2*]

**1** Xi'an Jiaotong University, School of Management, Xi'an, Shaanxi, China, **2** Shanxi Normal University, College of Mathematics and Computer Science, Taiyuan, Shanxi, China

\* renjingl@mail.xjtu.edu.cn (RL); xjs19640408@163.com (JX)

## Abstract

Under the federated learning framework, frequent parameter interactions between edge devices and servers result in communication inefficiency, while conventional encryption methods fail to resist multi-node collusion attacks. To address these challenges, this paper proposes an optimized federated learning scheme integrating adaptive channel pruning with multi-key homomorphic encryption. First, we construct a dynamic threshold determination mechanism that automatically calibrates channel pruning rates through precision feedback during the pre-pruning phase, achieving the optimal balance between model compression and accuracy, while significantly reducing communication bandwidth consumption compared to traditional algorithms. Second, based on the Brakerski-Gentry-Vaikuntanathan (BGV) multi-key fully homomorphic encryption architecture, we design a distributed public-key encryption protocol that enables aggregation servers to securely fuse multi-source model parameters without decryption, resisting collusion attacks from up to $C-1$ nodes (where $C$ denotes the total number of devices). Experiments on MNIST and CIFAR-10 datasets demonstrate that our scheme significantly reduces communication overhead through two complementary mechanisms: adaptive pruning reduces both the computational burden of local training and the volume of parameters transmitted per round, while multi-key BGV encryption ensures privacy-preserving aggregation without decryption. This work provides a novel technical pathway for privacy-preserving federated learning in resource-constrained scenarios.

## Introduction

In federated learning, edge devices need to engage in regular model parameter exchanges with the central server to achieve collaborative training. However, as machine learning models continue to grow in scale, communication and computational overhead issues become increasingly prominent. More complex models require greater bandwidth and computational resources to transmit and process parameters,

**Data availability statement:** All MNIST and CIFAR-10 dataset files used in this study are publicly available from established repositories: - The MNIST dataset of handwritten digits is available from Kaggle (https://www.kaggle.com/datasets/oddrationale/mnist-in-csv) and also accessible via TensorFlow Datasets (https://www.tensorflow.org/datasets/catalog/mnist). - The CIFAR-10 dataset is available from the official University of Toronto repository (https://www.cs.toronto.edu/~kriz/cifar.html) and through TensorFlow Datasets (https://www.tensorflow.org/datasets/catalog/cifar10).

**Funding:** This work was supported by the Fundamental Research Program of Shanxi Province (Grant No. 20210302124257). The funders provided financial support for this study but had no role in study design, data collection and analysis, decision to publish, or preparation of the manuscript.

**Competing interests:** The authors have declared that no competing interests exist.

while edge devices are typically resource-constrained, lacking sufficient bandwidth and processing power. To address these challenges, researchers have introduced model pruning techniques [1,2] in federated learning, effectively reducing the computational burden and communication overhead of edge devices. For example, Jiang et al. [3] proposed reducing communication costs by pruning weights; while Munir et al. [4] successfully shortened overall training time by pruning global models for lower-performing devices.

These works primarily focus on applying unstructured pruning techniques within the federated learning framework. Although such methods can maintain high accuracy even at high compression rates, they rely on specialized computation libraries and hardware support, making them unfavorable for deployment on resource-constrained edge devices. Xu et al. [5] effectively reduced communication overhead by employing structured channel pruning to decrease model size, enabling pruned models to be trained directly using existing hardware and computation libraries. However, these studies generally adopt a one-time pruning strategy, performing only a single pruning operation on the model. In this process, selecting an appropriate pruning rate is crucial: if the rate is too low, the model retains significant redundancy; if too high, it may result in considerable accuracy loss.

Furthermore, recent research [6] has revealed significant privacy and security challenges in federated learning. These challenges include: inference attacks [7], where malicious participants analyze model parameters or gradient information to deduce training data from specific clients; model reverse engineering [8], where attackers use reverse engineering techniques to reconstruct local models or approximate original data, leading to privacy breaches; and model extraction attacks [9], where malicious actors may extract partial data samples from the global model to understand data characteristics and distribution, ultimately resulting in privacy leakage. To counter these threats, researchers have proposed various privacy protection techniques, primarily including differential privacy (DP) [10], homomorphic encryption (HE) [11], and secure multi-party computation (SMC) [12]. However, these methods still face numerous challenges: federated learning based on differential privacy requires striking a delicate balance between model accuracy and privacy protection, constituting a significant challenge; while federated learning based on secure multi-party computation typically demands multiple rounds of interaction between participants to achieve secure aggregation, resulting in substantial communication overhead.

In comparison, federated learning approaches based on homomorphic encryption, despite certain limitations in computational efficiency, effectively avoid model accuracy degradation and complex interactions between clients, while achieving relatively ideal privacy protection outcomes. However, current homomorphic encryption-based federated learning schemes predominantly employ a single key for encrypting and homomorphically computing model parameters, meaning all ciphertexts involved in computation correspond to the same key. This design cannot effectively resist data leakage attacks from curious internal devices, nor collusion attacks between internal devices and servers. At present, although several federated learning-related studies

have separately focused on addressing privacy leakage [13] or communication overhead [14] issues, research that comprehensively tackles both challenges simultaneously remains relatively scarce.

Therefore, this research proposes an Adaptive Pruning Multi-Key Federated Learning (APMKFL) scheme based on the federated learning framework. In the pruning process of this scheme, edge devices apply various pruning rates for model pre-pruning and evaluate model accuracy on validation datasets. Subsequently, pre-pruned models are ranked according to their accuracy, and the pruning rate corresponding to the highest accuracy is automatically selected for final pruning operations. While network slimming [15] and multi-key BGV homomorphic encryption [16] are individually established techniques, their integration within a federated learning framework—with a feedback-driven adaptive threshold mechanism—constitutes the core novelty of this work. Three aspects distinguish APMKFL from a straightforward combination of prior methods: (1) Prior federated pruning works [1,3,5,17,18] apply static or one-time pruning. Network slimming [15] was designed for centralized training. We reformulate its BN-based channel scoring as a per-round adaptive decision mechanism within the FL communication protocol, where the pruning structure evolves with each gradient update—a non-trivial integration challenge not addressed by existing work. (2) Prior HE-based FL schemes [11, 12] use a single shared key, enabling honest-but-curious servers to decrypt individual updates. APMKFL employs multi-key BGV [16] with threshold decryption, so the server never holds a decryption key—achieving provable C-1 collusion resistance. (3) Unlike [19], which addresses both communication and privacy but uses differential privacy (degrading accuracy), APMKFL achieves communication efficiency and cryptographic privacy simultaneously with no accuracy trade-off. The main contributions of this research are as follows:

1) Proposed an adaptive iterative channel pruning method, enabling edge devices to dynamically adjust pruning thresholds based on the accuracy of pre-pruned models, thereby achieving optimal balance between model structural complexity and prediction accuracy.

2) Employed multi-key BGV homomorphic encryption technology, allowing edge devices to encrypt model parameters through jointly generated aggregation public keys, enabling servers to perform secure aggregation of local models from edge devices in ciphertext state to update the global model.

3) Conducted comprehensive experimental validation on standard MNIST and CIFAR-10 datasets, with results demonstrating that the proposed scheme significantly reduces communication overhead while maintaining high model accuracy, further enhancing data privacy and security protection.

## Related works

This section discusses related research works based on two major challenges faced by federated learning: high communication overhead and data privacy security.

Addressing the limited computational and communication resources of edge devices in federated learning, an increasing number of studies have incorporated model compression techniques into the federated learning framework. Major model compression methods include model pruning [20], knowledge distillation [21], and parameter quantization [22]. Jeong et al. [23] proposed federated distillation, utilizing knowledge distillation to transfer knowledge from large teacher models to smaller student models in federated learning environments. Prakash et al. [24] suggested using parameter quantization techniques to reduce parameter bit-width representation, thereby effectively decreasing communication overhead. This paper primarily focuses on techniques combining federated learning with model pruning, achieving greater efficiency by directly pruning the original model. Model pruning aims to remove redundant neurons, weights, or connections to reduce model size, lower computational and storage costs, while minimizing accuracy loss. For instance, Caldas et al. [1] employed lossy compression in federated learning to compress model storage footprint, reducing communication burden between edge devices and central servers by disabling sets of low-importance parameters. Yu et al. [17] proposed

using gating networks on edge devices to eliminate redundant neurons. Vahidian et al. [18] introduced a method combining structured and unstructured pruning approaches. However, these works generally adopt one-time pruning strategies. Research by Frankle et al. [25] demonstrated that iterative pruning approaches significantly reduce model accuracy loss compared to one-time pruning. Therefore, this paper conducts research from the perspective of iterative structural pruning.

Additionally, federated learning faces significant challenges in data privacy security. Within the federated learning framework, edge devices share only partial model parameters or gradients with the central server, rather than raw data. However, attackers may still exploit this shared information to infer private data from specific edge devices. To enhance data privacy protection in federated learning, researchers have proposed various solutions: Wei et al. [26] introduced a federated learning scheme incorporating differential privacy techniques, achieving edge device data privacy protection at the cost of significantly reduced model performance; Fang et al. [12] proposed using additive homomorphic encryption to protect model updates through the Paillier cryptosystem for privacy-preserving federated learning; Vedaraj et al. [27] presented a decentralized system that conducts secure statistical analysis on distributed datasets by applying the ElGamal elliptic curve additive homomorphic cryptosystem. Notably, in the aforementioned works, all participating edge devices utilize the same encryption and decryption keys. This design poses potential risks—private data information may leak between edge devices. More seriously, any curious edge device colluding with the server would compromise the data privacy of other edge devices.

Recently, some research has begun to simultaneously address the dual challenges of data privacy security and communication overhead in federated learning. Hu et al. [19] proposed an innovative approach that reduces communication rounds through periodic averaging while integrating secure aggregation with differential privacy techniques to effectively prevent data leakage. Drawing on the advantages of the method proposed by Hu et al. [19], our research scheme aims to tackle both fronts: reducing the communication frequency between edge devices and servers while decreasing the volume of transmitted data, thereby comprehensively lowering communication overhead while providing robust privacy protection mechanisms.

Reviewing existing works, we identify four critical gaps: (1) Pruning-only methods reduce communication but offer no privacy guarantees. (2) Single-key HE schemes protect privacy but are vulnerable to collusion and do not reduce communication volume. (3) Differential privacy methods sacrifice model accuracy due to noise injection. (4) Existing joint approaches either rely on trusted server-side data or inherit DP's accuracy penalty. APMKFL is the first scheme to jointly address all three objectives—communication efficiency, collusion-resistant privacy, and accuracy preservation—without trusted server data or noise-based privacy.

## Adaptive pruning multi-key federated learning framework

This section provides a detailed exposition of the proposed framework. First, it systematically introduces the technical principles and implementation process of the adaptive iterative channel pruning method. Next, it thoroughly analyzes the core mechanisms of the multi-key BGV homomorphic encryption method. Finally, it comprehensively presents the complete process and key stages of the joint model training performed by edge devices.

### Adaptive iterative channel pruning

A natural baseline is to apply iterative pruning with a fixed rate $\alpha$ throughout all communication rounds. However, this approach faces a fundamental dilemma: (1) If $\alpha$ is set too conservatively, communication savings are minimal. (2) If $\alpha$ is set aggressively, accuracy degrades significantly as the number of edge devices $C$ increases, because higher $C$ introduces more statistical heterogeneity, requiring greater model capacity to accommodate the distributional diversity across devices. (3) The optimal $\alpha$ differs across datasets, models, communication rounds, and device counts—making a universally good fixed choice impossible without per-experiment manual tuning. Our adaptive mechanism resolves this

by automatically identifying the highest pruning rate satisfying the accuracy constraint in each round, eliminating manual selection entirely. As illustrated in Fig 1, this paper investigates a federated learning architecture comprising a central server **S** and a set of edge devices **C** = $\{c_1, c_2, ..., c_n\}$. Each device $c \in$ **C** maintains a private local dataset $\mathcal{D}_c$ with $|\mathcal{D}_c|$ data samples. The global dataset is defined as $\mathcal{D} = \bigcup_{c \in \mathbf{C}} \mathcal{D}_c$, containing a total of $N = \sum_{c \in \mathbf{C}} |\mathcal{D}_c|$ samples. Under the standard federated learning framework, the central server **S** periodically aggregates local model parameters from all edge devices and learns the global model parameters $w$ by minimizing the global empirical risk:

$$\arg \min_{w} \mathcal{F}(w) = \sum_{c=1}^{n} \frac{|\mathcal{D}_c|}{N} f_c(w_c) \tag{1}$$

where $\mathcal{F}(w)$ denotes the global empirical risk function defined over the entire federated system, and $f_c(w_c)$ represents the local objective function with parameters $w_c$ specific to device $c$.

For each edge device $c$, the accuracy constraint for channel pruning is formally defined as follows: Given an $n$-layer CNN model with parameters $w_c$ and original channel configuration $\mathcal{L} = \{l_1, l_2, \ldots, l_n\}$ where $l_i$ denotes the number of output channels in the $i$-th layer, the channel pruning aims to find an optimized channel configuration $\mathcal{L}' \subseteq \mathcal{L}$ such that the pruned compact model $M_{\text{Pruned}}(\mathcal{L}', w_c')$ satisfies the target accuracy threshold $Acc_g$ on the validation set $\mathcal{D}_{\text{val}}$:

$$\max_{\mathcal{L}'} \left( M_{\text{Pruned}}(\mathcal{L}', w_c'), \mathcal{D}_{\text{val}} \right) \geq Acc_g \tag{2}$$

where $w_c'$ represents the parameters of the pruned model $M_{\text{Pruned}}(\mathcal{L}', w_c')$, and $Acc_g$ is the predefined accuracy threshold. The core objective of model pruning can be formalized as a constrained optimization problem: to find the optimal channel configuration that minimizes the model's structural complexity while satisfying the accuracy constraint.

In federated learning, the essence of model pruning lies in generating a mask structural binary matrix $mask_c$ for each edge device $c$, formally defined as:

$$w_c^* = w_c \odot mask_c \quad , mask_c \in \{0, 1\}^{|w_c|} \tag{3}$$

where $w_c^*$ denotes the sparsified model parameters after pruning, and $\odot$ represents the Hadamard product (element-wise multiplication).

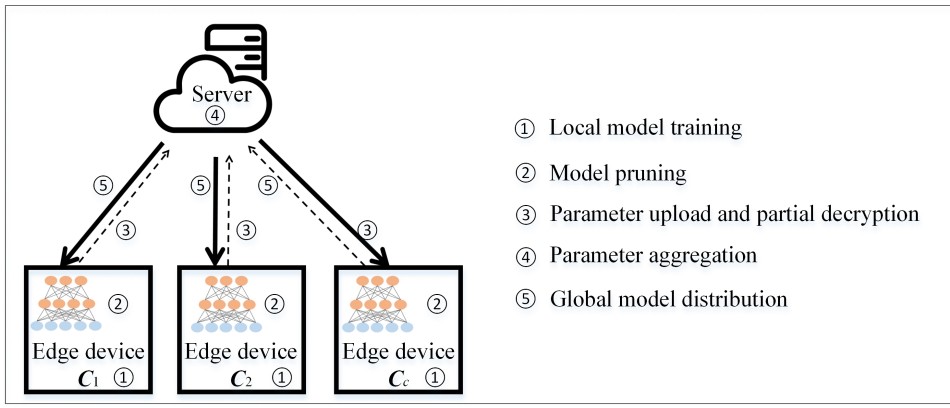

**Fig 1. The overview of system model.**

During the model pruning phase, the proposed method incorporates the channel pruning strategy presented in [15] by applying an $L_1$ regularization constraint to the scaling factor $\gamma$ of the batch normalization layer. This allows for dynamic identification of non-critical channels during training, generating a corresponding binary mask matrix where each element indicates the presence or absence of the respective channel rather than directly representing parameter pruning. Within the federated learning framework, each edge device trains a local model using its private local data and engages in a total of $T$ communication rounds with the central server for model aggregation.

Algorithm 1 proposes an adaptive threshold-based iterative channel pruning framework, which consists of two phases:

1) **Local Model Training** (Lines 1–3): During the $t$-th federated communication round, each edge device trains local model parameters $w_c^t$ on its private dataset $\mathcal{D}_c$, while simultaneously evaluating the model's validation accuracy $Acc_t$.

2) **Dynamic Channel Pruning** (Lines 4–13): When $Acc_t$ meet predefined accuracy constraints $Acc_g$, the pruning process proceeds through the following steps: Candidate pruning ratios $\alpha \in [0, 1]$ are uniformly sampled. During pre-pruning operations, the system first performs a complete sorting of scaling factors $\gamma$ in batch normalization layers. A pruning threshold is then calculated as $y[(\alpha \times \gamma_{sum})]$, where $\gamma_{sum}$ denotes the total number of scaling factors. A binary mask matrix $mask_c^\alpha$ is generated by comparing each channel's $\gamma$ value against the threshold: channels exceeding the threshold are preserved (marked as 1) while others are pruned (marked as 0). Each candidate pruning ratio $\alpha$ undergoes validation set evaluation, with the corresponding inference accuracy recorded as $Acc_\alpha$. Ultimately, the system selects the pruning configuration with the highest validation accuracy, determining the optimal mask matrix $mask_c^\alpha$ and model architecture.

**Algorithm 1. Adaptive Iterative Channel Pruning Algorithm.**

```
1:  for  t = 1, 2, ..., T do
2:      Train model on local dataset 𝒟_c:  w_c^t ← Train(𝒟_c)
3:      Calculate current model accuracy:  Acc_t ← Evaluate(w_c^t)
4:      if  Acc_g ≤ Acc_t  then
5:         (y, i) ← Sort(γ)                    ▷(y as the value of γ, i: index)
6:         for select pruning rate  α ∈ {α_1, α_2, ...}  do
7:             Calculate threshold:  thre ← y[(α × γ_sum)]
8:             Generate mask matrix:  mask_c^α = { 1,  γ > thre
                                                   { 0,  otherwise
9:             Pre-pruning:  w_c^* ← w_c^t ⊙ mask_c^α
10:            Evaluate pruned model:  Acc_α ← Evaluate(w_c^*)
11:         end for
12:         Select maximum accuracy:  MaxAcc_α ← max({Acc_α})
13:         Record optimal mask:  mask_c^α ← argmax_{mask_c^α} Acc_α
14:         Save pruned model:  w_c^* ← argmax_{w_c^*} Acc_α
15:      else
16:         continue
17:      end if
18:  end for
```

It is important to clarify that the candidate pruning rate selection in Algorithm 1 is fundamentally different from a conventional offline hyperparameter search. In a standard hyperparameter search, a fixed set of candidate values is evaluated on a held-out validation set before training begins, and the best value is then applied statically throughout the entire training process. In contrast, our adaptive mechanism operates online, within each communication round, and its behavior changes dynamically as the model evolves. Specifically, the pruning threshold is not chosen from a pre-fixed discrete grid; rather, it is derived from the current model's batch normalization scaling factors $\gamma$ via sorted ranking. Since the distribution of $\gamma$ changes with every gradient update, the set of channels identified for pruning at a given rate $\alpha$ is unique to each round. This means that the same nominal rate $\alpha$ can preserve entirely different subsets of channels in round t versus

round t + 1. Therefore, the mechanism is self-calibrating: the accuracy feedback in each round directly shapes the effective pruning structure, a property that offline hyperparameter search cannot replicate. Table 1 summarizes the key distinctions.

The extra computational cost introduced by the adaptive pruning decision in each communication round consists of two components. First, sorting the BN scaling factors $\gamma$ incurs $O(|\gamma| \log |\gamma|)$ time, where $|\gamma|$ denotes the total number of channels across all layers (a one-time operation per round regardless of $k$). Second, for each of the $k$ candidate pruning rates, the algorithm performs one forward pass over the local validation set $D_{\text{val}}$ to compute the pruned model's accuracy. Each forward pass costs $O(|D_{\text{val}}| \cdot \text{FLOPs}(\mathbf{w}_c^*))$, where $\text{FLOPs}(\mathbf{w}_c^*)$ is the inference cost of the pruned model—which is smaller than the full model. The total additional cost per round is therefore $\boldsymbol{O(|\gamma| \log |\gamma| + k \cdot |D_{\text{val}}| \cdot \text{FLOPs}(\mathbf{w}_c^*))}$.

## Multi-key BGV homomorphic encryption mechanism

Chen et al. [16] proposed the first BGV-based multi-key fully homomorphic encryption (MKFHE) methods, which operates on ring elements and derives its security from the ring learning with errors (RLWE) problem. This multi-key homomorphic encryption mechanism allows distinct edge devices to perform encryption using individual private keys, while requiring collaborative participation from all involved devices during decryption. In our federated learning framework, we specifically leverage the additive homomorphic property of MKFHE. By aggregating public keys from all edge devices to form a unified public key, the scheme achieves secure parameter aggregation. The specific implementation details of the sub-algorithm are described as follows:

1) **Initialization Phase:** Given a security parameter $\lambda$ and an edge device set $\mathcal{C}$, the system establishes cryptographic parameters through the following steps: I) Algebraic Structure Definition: Construct the cyclotomic polynomial ring $R = \mathbb{Z}[X]/(X^n + 1)$ with polynomial dimension $n$ being a power of two, ensuring compatibility with NTT (Number Theoretic Transform) computations. II) Modulus Selection: Choose coprime integers $q$ (ciphertext modulus) and $p$ (plaintext modulus) such that $p \ll q$, which guarantees effective noise control in homomorphic operations. III) Noise Distribution: Define a bounded discrete Gaussian distribution $\chi$ over $R$ with noise bound $B$, governing the statistical properties of encryption noise terms. IV) Public Parameter Generation: Randomly sample a public vector $a \leftarrow R_q$, where the quotient ring $R_q$ is constructed via $R/qR \cong \mathbb{Z}_q[X]/(X^n + 1)$. The complete public parameters are $pp = \{R, \chi, B, q, a, p\}$.

2) **Key Generation Phase:** During the key generation process for edge device $c$, the core private parameter $z_c \leftarrow R_3$ is first randomly selected from the ternary polynomial ring $R_3 = \mathbb{Z}[x]/(x^n + 1, 3)$, forming a computationally efficient private key $sk_c = s_c := (1, -z_c) \in R_3^2$. This structure fixes the leading coefficient as 1 to eliminate modular reduction in polynomial multiplication, thereby significantly reducing computational overhead. Next, a noise term $e_c \leftarrow \chi$ is sampled from a bounded noise distribution $\chi$, and combined with the predefined public parameter $a$ to generate the public key

**Table 1. Comparison with conventional hyperparameter search.**

| Aspect | Conventional Hyperparameter Search | Our Adaptive Mechanism |
|---|---|---|
| Execution Timing | Before training, offline | During each training round, online |
| Candidate Source | Manually predefined discrete grid | Dynamically generated by sorted ranking of BN layer's $\gamma$ |
| Pruning Structure | Fixed (independent of rounds) | Varies per round (evolves with the $\gamma$ distribution) |
| Accuracy Feedback | Used only to select the optimal hyperparameter | Directly determines the final pruning mask for the current round |
| Adaptivity | No (remains fixed after selection) | Yes (adapts automatically as the model converges) |
| Computational Cost | One-time offline cost | Incremental per-round cost |

component $b_c = az_c + pe_c \bmod q$. The public key is then defined as $pk_c = (b_c, a) \in R_q^2$, whose security relies on the hardness assumption of the RLWE problem. The resulting key pair $\{pk_c, sk_c\}$ guarantees ciphertext consistency, where the public key is used for data encryption and the private key exclusively serves decryption purposes.

3) **Encryption Phase:** To encrypt a plaintext $\mu \in R_p = \mathbb{Z}[x]/(x^n + 1, p)$ using the public key $pk_c = (b_c, a) \in R_q^2$ of edge device $c$, first sample a random polynomial $r_c \leftarrow R_2 = \mathbb{Z}[x]/(x^n + 1, 2)$ and independently draw noise terms $e_0, e_1 \leftarrow \chi$ from a bounded noise distribution. The ciphertext $ct_c = (c^0, c^1) \in R_q^2$ is computed as:

$$\begin{cases} c^0 = r_c \cdot b_c + p \cdot e_0 + \mu \bmod q \\ c^1 = r_c \cdot a + p \cdot e_1 \bmod q \end{cases}$$

(4)

This encryption employs a dual-masking mechanism (via $r_c$ and $e_0$, $e_1$) to ensure semantic security, with its safety reduced to the hardness assumption of the RLWE problem.

4) **Decryption Phase:** Upon receiving ciphertext $ct_c = (c^0, c^1)$, edge device $c$ computes the inner product with private key $sk_c = (1, -z_c)$:

$$\langle ct_c, sk_c \rangle = c^0 - z_c \cdot c^1 \bmod q = p \cdot \underbrace{(e_0 - z_c e_1)}_{e} + \mu \bmod q$$

(5)

When the noise term $e$ satisfies $\|p \cdot e\|_\infty < q/2$, the plaintext is accurately recovered via modulo-$p$ operation:

$$\mu = \left( \langle ct_c, sk_c \rangle \bmod q \right) \bmod p$$

(6)

The co-design of noise scaling factor $p$ and parameter $q \gg p$ guarantees decryption robustness.

5) **Homomorphic Addition:** Let $ct_i = (c_i^0, c_i^1)$ and $ct_j = (c_j^0, c_j^1)$ be ciphertexts encrypting $\mu_i, \mu_j \in R_p$ for devices $i$ and $j$, respectively. To perform homomorphic addition, construct an extended ciphertext $\overline{ct} = \left( c_i^0 + c_j^0, \ c_i^1, \ c_j^1 \right) \in R_q^{C+1}$ and generate an extended private key $\overline{sk} = (1, -z_i, -z_j) \in R_3^{C+1}$, satisfying linear homomorphism:

$$\langle \overline{ct}, \overline{sk} \rangle = \underbrace{(c_i^0 + c_j^0)}_{\text{plaintext sum}} - z_i c_i^1 - z_j c_j^1 \equiv p \cdot (\tilde{e}_i + \tilde{e}_j) + (\mu_i + \mu_j) \bmod q$$

(7)

The plaintext sum $\mu_i + \mu_j$ is obtained via $\bmod p$ operation, enabling ciphertext arithmetic without decryption.

6) **Threshold Decryption Mechanism:** The proposed scheme supports threshold decryption, enabling collaborative decryption with partial private keys. The aggregated ciphertext after homomorphic operations takes the form $\bar{ct} \in R_q^{C+1}$. The concrete procedure comprises:

I) **Partial Decryption:** Each edge device $c$ samples random noise $e_c^{sm}$ from the noise distribution $\psi$, then computes the local decryption share using its private key component:

$$em_c = c_c^1 \cdot (-z_c) + e_c^{sm} \bmod q$$

(8)

Here, $e_c^{sm}$ serves to protect the confidentiality of $z_c$ against potential leakage.

 

II) **Decryption Fusion:** Aggregate all local decryption shares $\{em_1, ..., em_C\}$ and recover the plaintext via two-step modular reduction:

$$\mu_{sum} = \sum_{c=1}^{C} c_c^0 + \sum_{c=1}^{C} em_c (\bmod\ q\ \bmod\ p) \tag{9}$$

## Framework implementation and workflow

Inspired by the seminal work of Moore et al. [28], the proposed federated learning framework in this study represents a significant extension of the Federated Averaging (FedAvg) methodology. Within the system model, the total communication rounds between edge devices and the central server are specified as $T$ iterations, with the collaborating edge device cohort formally defined as $\mathbf{C} = \{c_1, c_2, ..., c_n\}$. As illustrated in Fig 1, the architectural design comprises seven critical operational phases, where Algorithm 2 provides a comprehensive procedural breakdown of the joint model training mechanism.

(1) **Parameter Initialization Protocol:** The central server initializes and broadcasts cryptographic public parameters $pp$. Each edge device generates distinct cryptographic key pairs $\{pk_c, sk_c\}$ based on $pp$. Subsequently, all edge devices collaboratively compute the aggregated public key through a secure multi-party computation protocol:

$$\bar{b} = \sum_{c=1}^{C} b_c = \sum_{c=1}^{C} az_c + \sum_{c=1}^{C} pe_c \ (\bmod\ q) \tag{10}$$

(2) **Privacy-Enhanced Local Training** During the initialization phase of the t-th federated learning communication round, each edge node synchronizes the global model parameters $W^{(t-1)}$ through a secure parameter channel from the central coordinator. Utilizing local non-IID data $\mathcal{D}_c$, every node implements differentially private SGD with adaptive gradient clipping (DP-AC-SGD) for E complete training epochs:

$$w_c^{(t)} = w_c^{(t-1)} - \eta_t \left[ \frac{1}{|\mathcal{B}|} \sum_{\xi \in \mathcal{B}} \nabla \mathcal{L}(\xi; w_c^{(t-1)}) \right] \cdot \min\left(1, \frac{\gamma}{\|\nabla \mathcal{L}\|_2}\right) \tag{11}$$

where $\mathcal{B}$ denotes the randomly sampled data batch and $\gamma$ represents the dynamic gradient clipping threshold. After training, nodes perform adaptive pruning on parameters $w_c^{(t)}$ according to Algorithm 1.

(3) **Model Parameter Encryption:** Building upon the ring-element encoding scheme proposed by Dowlin et al. [29], our method initiates with structured encoding of local model parameters. For any rational number $b$, its binary expansion is expressed as $b_{n_1} \cdots b_1 b_0 . b_{-1} \cdots b_{-n_2}$, where the integer part $b_{n_1} \cdots b_1 b_0$ contains $n_1 + 1$ significant digits, and the fractional part $b_{-1} \cdots b_{-n_2}$ maintains $n_2$-bit precision. The polynomial ring mapping mechanism defines the encoding formula as:

$$b = \sum_{i=0}^{n_1} b_i x^i + \sum_{i=0}^{n_2} (-b_{-i}) x^{n-i} \tag{12}$$

where $n$ denotes the dimension parameter of the polynomial ring. A representative example is the value 3.5 with binary expansion $(11.1)_2$, corresponding to the ring element representation $x + 1 - x^{n-1}$.

During the parameter encryption phase, edge devices employ the aggregated public key $\bar{b}$ to encrypt local model parameters $w_c^{(t)}$. The detailed procedure involves: randomly selecting parameters $r_c, e_c, e_c'$ from the noise distribution $\chi$, then computing the ciphertext pair:

$$ct_c = (c_0, c_1) = \left( r_c \bar{b} + pe_c + w_c^t, r_c a + pe_c' \right) \bmod q \tag{13}$$

Finally, edge devices transmit the encrypted result $ct_c$ to the central server.

(4) **Local Model Homomorphic Aggregation:** The central server leverages homomorphic accumulation properties to perform ciphertext-space aggregation on $C$ received edge device ciphertexts $ct_c = (c_c^0, c_c^1)$, generating the global model ciphertext:

$$ct_{\text{sum}} = \sum_{c=1}^{C} ct_c = \left( \sum_{c=1}^{C} c_c^0, \sum_{c=1}^{C} c_c^1 \right) = (c_{\text{sum}}^0, c_{\text{sum}}^1) \bmod q \tag{14}$$

(5) **Distributed Partial Decryption:** Each edge device $c$ performs partial decryption on the global ciphertext $ct_{\text{sum}}$ using its private key $z_c$, producing a decryption share:

$$em_c = \left( -z_c \cdot c_{\text{sum}}^1 + e_c^{\text{sm}} \right) \bmod q = \left( -z_c \cdot \sum_{c=1}^{C} c_c^1 + e_c^{\text{sm}} \right) \bmod q \tag{15}$$

(6) **Decryption Synthesis and Reconstruction:** After collecting all partial decryption results $\{em_c\}_{c=1}^{C}$, the central server executes decryption synthesis:

$$\sum_{c=1}^{C} w^{(t)} = c_{\text{sum}}^0 + \sum_{c=1}^{C} em_c \ (\bmod q) \tag{16}$$

This operation reconstructs the unencrypted aggregated model parameters.

(7) **Global Model Update:** The server decodes the aggregation result and computes the weighted average to generate the next-generation global model:

$$w^{(t+1)} = \frac{1}{C} \left( \sum_{c=1}^{C} w_c^{(t)} \right) \tag{17}$$

(8) **Termination Criteria:** The parameter initialization phase is executed only once before the first training round, with generated public parameters reused in subsequent iterations. The protocol terminates when either condition is met: 1) Global model loss function converges ($\|\nabla \mathcal{L}(w^{(t)})\| < \epsilon$); 2) Preset maximum iteration count is reached.

**Algorithm 2. Federated Training Model.**

**Input:** Edge datasets $\{\mathcal{D}_1, ..., \mathcal{D}_C\}$ where $\mathcal{D}_c$ is c-th device's data; Initial global parameters $W$; Edge device set $C$; Communication rounds $T$

```
Output: Converged edge models
  Server Protocol:
1:  for round t=1 to T do
2:      Activate devices C = {1,2,...,C}
3:      parallel for each c ∈ C:
4:      E(W_c^{t+1}), em_c ← ClientUpdate(c, W_c^t)
5:      Aggregate W^{t+1} = 1/C ∑_{c=1}^{C} W_c^{t+1}
6:  end for
  ClientUpdate(c, W_c^t):
7:  for epoch i=1 to E do
8:      for batch b ∈ B do
9:          Update with mask:
10:            W_c^{t+1} ← W_c^t ⊙ mask_c^t − η∇f_{(c)}(W_c^t ⊙ mask_c^t; b)
11:        end for
12:       Prune via Algorithm1: (W_c^{t+1}, mask_c^{t+1}) ← Algorithm1(W_c^{t+1})
13:       Encrypt params: E(W_c^{t+1}) ← Eq(13)(W_c^{t+1})
14: end for
15: return E(W_c^{t+1}), em_c
```

## Scheme analysis

### Security analysis

The proposed scheme in the federated learning scenario employs a multi-key homomorphic encryption method to safeguard data privacy, adhering to the security requirements of the semi-honest model. This implies that both the central server and all edge devices act honestly but remain curious. In other words, they strictly follow the protocol while attempting to infer private data of other devices from the shared information during protocol computation.

This section demonstrates the security of the proposed scheme from three perspectives.

**Theorem 1: Security against honest-but-curious central server.** An honest-but-curious central server cannot infer any private data from the edge devices.

**Proof:** In the APMKFL federated learning scheme, edge devices transmit two types of information to the server. First, in step 2, the edge devices send the ciphertext of local model parameters $ct_c$ to the central server, which is generated by multi-key BGV encryption. Then, in step 4, the edge devices send the partially decrypted global model result $em_c$ to the central server. The ciphertext and decryption result are expressed as follows:

$$ct_c = (c^0, c^1), \quad c^0 = r_c \cdot \bar{b} + pe_c + w_c' \pmod{q}$$

$$c^1 = r_c \cdot a + pe_c' \pmod{q}, \quad em_c = (-z_c) \cdot c_{sum1}^1 + e_c \pmod{q} \tag{18}$$

According to the RLWE assumption, all shared information contains an additional error term to guarantee security. The RLWE ensures that $c^0$ and $em_c$ are computationally indistinguishable from uniformly random elements of $R_q$. Therefore, these values do not disclose any information about the plaintext $w_c'$ or the key $(-z_c)$ to the central server. After performing the final decryption, the central server can only obtain the sum of the local model parameters from all edge devices without revealing any individual parameter.

Consequently, the proposed scheme can ensure the security of individual model parameters, effectively protecting the data privacy on edge devices. The central server cannot infer any private information of the edge devices from the received data.

**Theorem 2: Security against honest-but-curious edge devices.** An honest-but-curious edge device cannot infer any private data from other edge devices by eavesdropping on shared information.

**Proof:** In the APMKFL scheme, the model parameters of each edge device are encrypted using multi-key BGV encryption based on RLWE. Each edge device possesses its own public and private keys and collaboratively computes an aggregated public key to encrypt its model parameters. The decryption of the global model ciphertext requires all edge devices to compute their respective partial decryption results and send them to the central server for final decryption.

To ensure the security of edge devices' private keys, each edge device introduces an error term in its partial decryption result. This prevents private key leakage, ensuring that even an honest-but-curious edge device cannot infer any private information about another edge device's local data by intercepting the uploaded information.

**Theorem 3: Security against collusion between edge devices and the central server.** The proposed scheme is secure against collusion between the central server and up to $C-1$ edge devices, where $C$ denotes the total number of edge devices.

**Proof:** In the APMKFL scheme, each edge device encrypts its model parameters using the aggregated public key $\bar{b}$ before uploading them to the central server. The server then computes the sum of all edge devices' local model parameters. The ciphertexts $ct_i$ and the aggregated ciphertext $ct_{sum}$ can only be decrypted through collaborative partial decryption from all edge devices.

**Type-I collusion attack:** An edge device colludes with the central server to recover the plaintext model parameters $w_i^t$ from the ciphertext $ct_i$ of a compromised edge device $c_i$. In the worst-case scenario, $C-1$ edge devices collude with the central server, leaving only $c_i$ uncompromised. The colluding parties compute $c_i^1 \cdot (-z_j)$ for $j \neq i$ and combine with $c_i^0$:

$$c_i^0 + \sum_{j \neq i}^{C} c_i^0 \cdot (-z_j) = r_i \cdot \bar{b} + pe_c + w_i^t + \sum_{j \neq i}(r_i \cdot a + pe_i') \cdot (-z_j)$$

$$= r_i \cdot (-z_i) \cdot a + w_i^t + pe_c + \sum_{j \neq i} pe_i' \cdot (-z_j)$$

(19)

The result remains a partially encrypted ciphertext under the public key $b_i$ of the compromised device $c_i$. Even with access to the private keys $s_j$ of other edge devices, the colluding parties cannot decrypt $ct_i$ and thus cannot access any private information.

**Type-II collusion attack:** Edge devices and the central server attempt to infer a single local model $w_i^t$ from the decrypted global model $w^{t+1}$. The scheme ensures that such inference is impossible as long as at least two edge devices do not participate in the collusion. In the worst case, $C-2$ edge devices and the central server collude. By subtracting the known local models of these $C-2$ devices from the global model, the colluding parties can only obtain the sum of the remaining two devices' models, without identifying either individually.

Thus, the APMKFL scheme is resilient to collusion attacks involving up to $C-1$ edge devices and the central server, ensuring the privacy and security of local model data.

## Correctness analysis

This section analyzes and proves the correctness of the decryption process for the global model ciphertext in the proposed scheme.

**Theorem 4:** The global model ciphertext can be correctly decrypted with the collaboration of all edge devices.

**Proof:** After collecting the partial decryption results from all edge devices, the central server performs the final decryption as follows:

$$\sum_{c=1}^{C} w^t = c_{sum}^0 + \sum_{c=1}^{C} em_c \mod q$$

$$= c_{sum}^0 + \sum_{c=1}^{C}(-z_c) \cdot \sum_{c=1}^{C}(r_c a + pe_c') + \sum_{c=1}^{C} e_c^{sm} \mod q \mod p$$

$$= \sum_{c=1}^{C}(r_c \bar{b} + pe_0 + \mu_c) + \sum_{c=1}^{C}(-z_c) \cdot \sum_{c=1}^{C}(r_c a + pe_1) + \sum_{c=1}^{C} e_c^{sm} \mod q \mod p$$

$$= \sum_{c=1}^{C} r_c a z_c + \sum_{c=1}^{C} r_c pe_c + \sum_{c=1}^{C}(pe_0 + \mu_c) + \sum_{c=1}^{C}(-r_c a z_c)$$

$$+ \sum_{c=1}^{C}(-z_c pe_1 + e_c^{sm}) \mod q$$

$$= \sum_{c=1}^{C} \mu_c + \sum_{c=1}^{C}(r_c pe_c + pe_0 - z_c pe_1 + e_c^{sm}) \mod q \mod p$$

$$= \sum_{c=1}^{C} \mu_c + \sum_{c=1}^{C} p(r_c e_c + e_0 - z_c e_1) + e_c^{sm} \mod q \mod p$$

$$= \sum_{c=1}^{C} \mu_c$$

$$(20)$$

Hence, the final decryption result is the sum of the plaintext model parameters from all edge devices, which constitutes the global model. This completes the proof of correctness.

## Experiments and results analysis

This section first provides a brief overview of the experimental setup. Then, it analyzes the model size after iterative pruning under different precision constraints in the APMKFL scheme and examines the impact of pruning on communication overhead. Finally, the performance of APMKFL is evaluated through comparisons with four popular federated learning schemes.

### Experimental setup

This experiment was conducted on a Windows 11 operating system, using an Intel i7-12700F processor, GTX 3060Ti GPU, and 8GB RAM. All neural network models were built using Python's PyTorch framework. We evaluated the performance of APMKFL on two classic image recognition tasks: MNIST digit recognition and CIFAR-10 image classification. The MNIST dataset consists of 10 classes, comprising 60,000 training images and 10,000 test images, with each image being a 28×28 pixel grayscale image. The CIFAR-10 dataset also contains 10 classes, comprising 50,000 training images and 10,000 test images, with each image being a 32×32 pixel color image.

For IID experiments, all training samples are randomly shuffled and uniformly distributed across $C$ edge devices, such that each device holds an approximately equal number of samples from all classes. For Non-IID experiments, we adopt the Dirichlet distribution-based partitioning strategy, which is a widely used and principled method for simulating heterogeneous data in federated learning [30]. Specifically, for each class $k$, the proportion of samples allocated to device $c$ is drawn from a Dirichlet distribution $\text{Dir}(\alpha)$, where $\alpha$ is the concentration parameter controlling the degree of heterogeneity: smaller $\alpha$ induces more severe Non-IID distribution (fewer classes per device), while $\alpha \to \infty$ approaches the IID case. In

this study, we set $\alpha$ = 0.5, a standard value in the federated learning literature that produces moderate-to-strong non-IID conditions. In this setting, each edge device typically receives samples dominated by 1–3 classes. Table 2 below summarizes the resulting data distribution characteristics for each experimental configuration.

In the federated learning system, we configured experiments with 10, 20, and 30 edge devices for comparative analysis. Different neural network architectures were employed for different datasets: a simple network consisting of two convolutional layers and two fully connected layers (2NN) was used for the MNIST dataset, while the more complex VGG11 architecture was adopted for the CIFAR-10 dataset. The experimental parameter configurations are detailed in Table 3.

All edge devices utilized the Stochastic Gradient Descent algorithm for local model training. The specific parameter settings were as follows: 50 local iterations (Epochs), 20 total communication rounds ($k$) with the central server, a mini-batch size of 64, and an initial learning rate ($\eta$) of 0.01. Additionally, the selection of security parameters for multi-key homomorphic encryption required balancing efficiency and security. In this experiment, all edge devices shared global parameters (N, q, and p), with each client generating a unique public-private key pair based on these parameters. To ensure a 128-bit security strength, we set the security parameters as N = 4096, q = 218, and p = 128.

To ensure fair comparison, all methods share identical base hyperparameters. Method-specific parameters follow their respective original papers, as detailed in Table 4.

## Performance evaluation

The performance of the APMKFL scheme is evaluated on the MNIST and CIFAR-10 datasets under both IID and non-IID settings, focusing on the model size after iterative pruning under different accuracy constraints, i.e., the model pruning rate. The accuracy constraints are set to 90%, 85%, and 80%, with the number of edge devices fixed at 10. The pruning rate variation at each edge device is recorded under different accuracy levels. Fig 2 illustrates the variation in the number

**Table 2. Summary of experimental datasets and partitioning strategies.**

| Dataset | Train/Test Split | Devices (C) | Partition Method | Avg Samples/Device | Typical Distribution |
|---------|------------------|-------------|------------------|--------------------|--------------------|
| MNIST | 60,000 / 10,000 | 10 | Dirichlet ($\alpha$ = 0.5) | `600 / device` | ~2–3 classes dominant |
| CIFAR-10 | 50,000 / 10,000 | 10 | Dirichlet ($\alpha$ = 0.5) | `5,000 / device` | ~2–3 classes dominant |
| MNIST | 60,000 / 10,000 | 20 | Dirichlet ($\alpha$ = 0.5) | `300 / device` | ~1–2 classes dominant |
| CIFAR-10 | 50,000 / 10,000 | 20 | Dirichlet ($\alpha$ = 0.5) | `2,500 / device` | ~1–2 classes dominant |

**Table 3. Experimental setup.**

| Model | Model Parameters | Dataset | Number of Edge Devices |
|-------|------------------|---------|------------------------|
| 2NN | 0.08MB | MNIST | 10, 20, 30 |
| VGG11 | 35.2MB | CIFAR-10 | 10, 20, 30 |

**Table 4. Hyperparameter settings for all compared methods.**

| Method | Shared Hyperparams | Method-specific Params |
|--------|--------------------|------------------------|
| FedAvg [31] | SGD, $\eta$ = 0.01, $E$ = 50, $B$ = 64 | N/A |
| ESFL [32] | SGD, $\eta$ = 0.01, $E$ = 50, $B$ = 64 | Top-$k$ = 30% |
| CPFed [33] | SGD, $\eta$ = 0.01, $E$ = 50, $B$ = 64 | $\varepsilon$ = 1.0 (DP noise) |
| FedDUAP [34] | SGD, $\eta$ = 0.01, $E$ = 50, $B$ = 64 | Pruning ratio = 0.6 |
| APMKFL (Ours) | SGD, $\eta$ = 0.01, $E$ = 50, $B$ = 64 | $\alpha \in \{0.1, \ldots, 0.9\}$, $k$ = 9 |

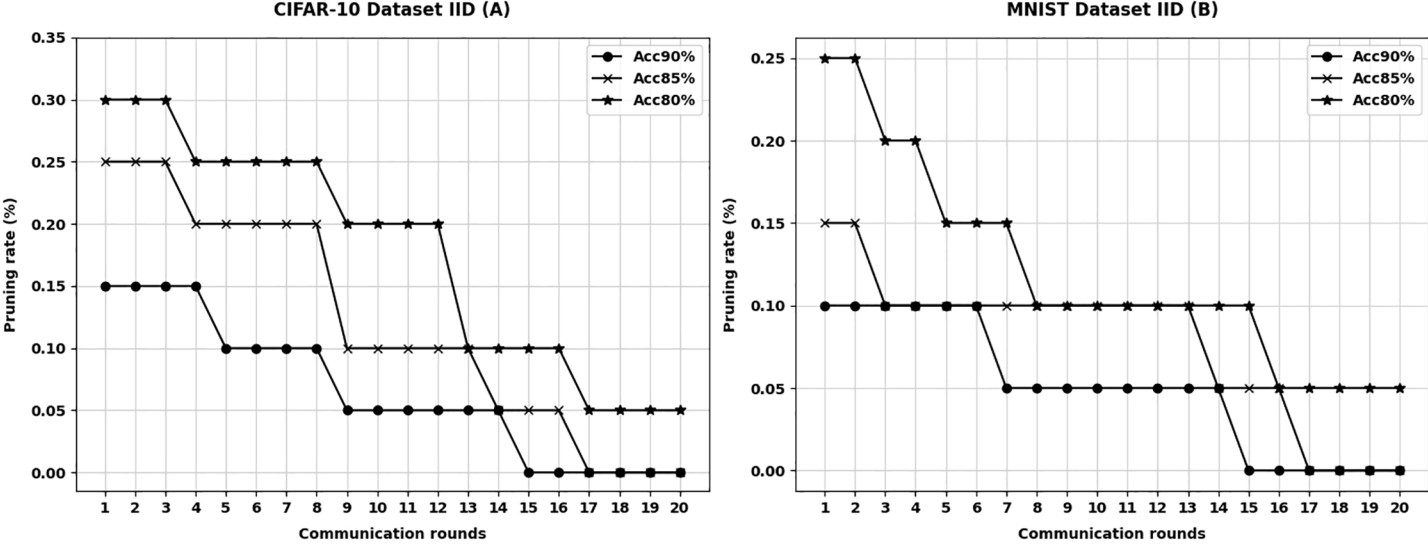

**Fig 2. IID settings for different datasets.**

of communication rounds and pruning rates for different datasets under the IID setting, while Fig 3 presents the corresponding results under the non-IID setting.

In both the MNIST and CIFAR-10 datasets, each edge device performs model pruning after local training and before communicating with the central server. At this stage, the model accuracy has already satisfied the predefined accuracy constraint, and the accuracy remains within the constraint even after pre-pruning. Each edge device selects the pruned model with the highest pruning rate that still satisfies the accuracy constraint for communication.

For IID data, as shown in Fig 2, after several rounds of iterative pruning, further pruning ceases, and the model accuracy gradually approaches the target constraint. Compared to the original unpruned model, the final pruning rates on the CIFAR-10 dataset are 79.2% (for 90% accuracy constraint), 86.4% (85%), and 92.8% (80%). On the MNIST dataset, the pruning rates are 60.2% (90%), 76.4% (85%), and 88.3% (80%), respectively. Moreover, the pruned models still satisfy the accuracy constraints, indicating that the 2NN and VGG11 models are overparameterized for the MNIST and CIFAR-10 classification tasks, respectively. Pruning redundant parameters does not degrade model accuracy; on the contrary, it can even improve accuracy by making the model structure more compact and better fitted to the data, thus enhancing training efficiency. Through iterative pruning, the model achieves a balance between communication cost and accuracy. Further pruning beyond this point would lead to performance degradation, as the model's capacity would be overly reduced and fail to meet the accuracy constraint.

For non-IID data, the accuracy of the 2NN and VGG11 models is generally lower than in the IID setting. As shown in Fig 3, the number of pruning iterations is also reduced accordingly. Compared to the original unpruned models, the model sizes are significantly reduced. Specifically, in the CIFAR-10 dataset, the pruning rates under different accuracy constraints are 65.3% (90%), 74.7% (85%), and 86.2% (80%). In the MNIST dataset, the pruning rates are 56.2% (90%), 72.4% (85%), and 89.3% (80%). Therefore, under the premise of satisfying the accuracy constraint, APMKFL effectively compresses the model through iterative pruning, making the model more compact and significantly reducing the communication overhead.

## Comparison with existing schemes

**Functional analysis.** To address the high communication overhead and privacy concerns in federated learning, many innovative approaches have emerged in recent years. The ESFL scheme [32] integrates Top-k gradient sparsification with

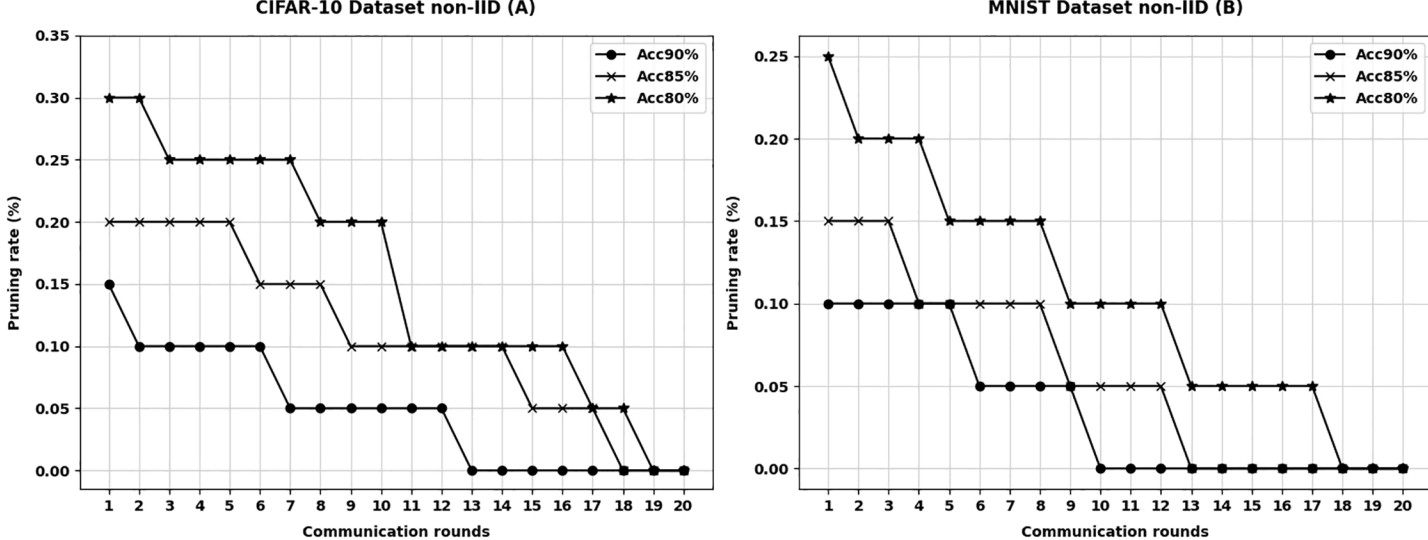

**Fig 3. Non-IID settings for different datasets.**

Paillier homomorphic encryption, allowing only partial gradients to be uploaded in each communication round, thereby reducing the communication overhead while ensuring gradient privacy through semi-homomorphic encryption. The CPFed scheme [33] combines sketch-based gradient compression with differential privacy, aiming to reduce transmission costs while enhancing privacy protection. The FedDUAP scheme [34] introduces a pruning strategy based on insensitive server-side data and decentralized edge device data to compress the global model and lower communication costs.
In addition, the classic FedAvg [31] method serves as a baseline for comparison. FedAvg does not incorporate model pruning or homomorphic encryption, making it less efficient in communication and lacking privacy-preserving capabilities. Table 5 presents a functional comparison across four aspects: accuracy retention, communication cost reduction, model parameter protection, and resistance to collusion attacks.

Specifically, ESFL [32] applies semi-homomorphic encryption to protect gradient privacy, but since all edge devices use a shared key, it is vulnerable to collusion between the server and edge devices. CPFed [33], on the other hand, achieves resistance to collusion through differential privacy. However, the inherent trade-off between privacy and model accuracy—caused by the noise added to the gradients—poses a significant challenge. FedDUAP [34] employs layer-wise model pruning to reduce communication overhead, but it assumes the availability of a portion of training data on the server side, thereby failing to ensure data privacy.

**Table 5. Function comparison of different schemes.**

| Scheme | Model Accuracy Preservation | Communication Reduction | Model Parameter Protection | Resistance to Collusion Attacks |
|---|---|---|---|---|
| FedAvg [31] | 51 | 55 | 55 | 55 |
| ESFL [32] | 51 | 51 | 51 | 55 |
| CPFed [33] | 55 | 51 | 51 | 51 |
| FedDUAP [34] | 51 | 51 | 55 | 55 |
| APMKFL | 51 | 51 | 51 | 51 |

**Comparison of model accuracy and communication overhead.** This section evaluates the performance of APMKFL in terms of model accuracy and communication overhead, in comparison with other approaches under varying numbers of edge devices. In federated learning, there exists a trade-off between model accuracy and communication cost—reducing communication overhead typically involves transmitting fewer parameters, which may adversely affect model performance. Figs 4 and 5 illustrate the comparison of model accuracy and communication overhead on the CIFAR-10 and MNIST datasets, respectively, under different edge device settings (C = 10, 20, 30).

Top-k sparsification selects only the top-k gradient elements for transmission, reducing communication overhead. While [35] suggests k has limited effect on convergence speed in homogeneous (IID) settings, the appropriate k under Non-IID federated learning is non-trivial. To empirically justify k = 30%, we conduct a sensitivity analysis over k ∈ 10%, 20%, 30%, 40% on CIFAR-10 under both IID and Non-IID settings. Results are summarized in Table 6.

(1) k = 10% achieves near-FedAvg accuracy but yields only 10% communication reduction—insufficient for practical deployment. (2) k = 20% provides a favorable accuracy-communication trade-off but is slightly sensitive to device count under Non-IID. (3) k = 30% delivers a consistent 70% communication reduction with acceptable accuracy across all *C* values under both IID and Non-IID settings, corroborating [35]'s recommendation. (4) k = 40% offers only marginal accuracy

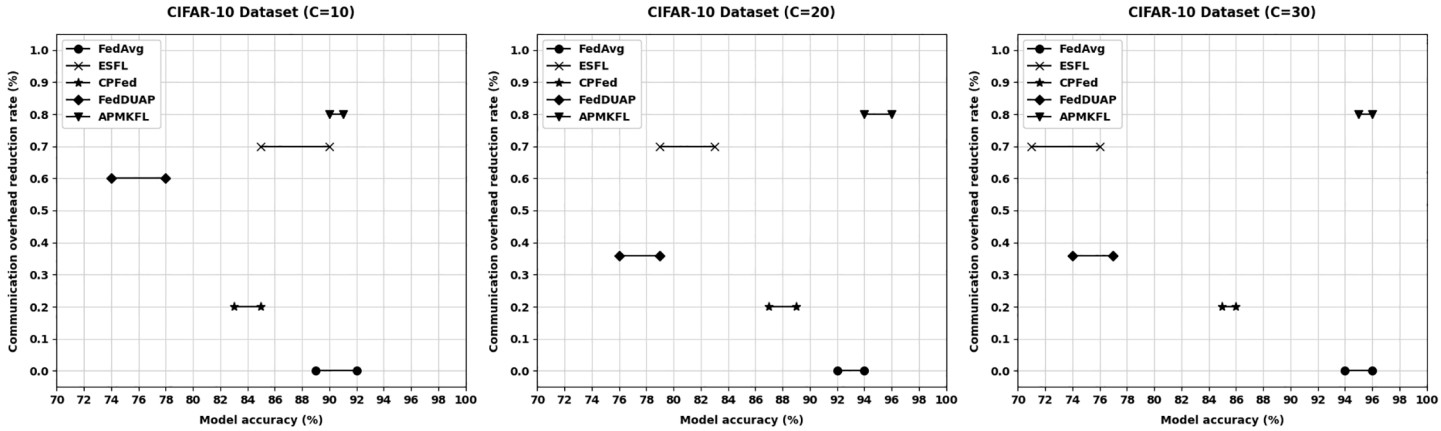

**Fig 4. Comparison of different schemes (CIFAR-10 dataset).**

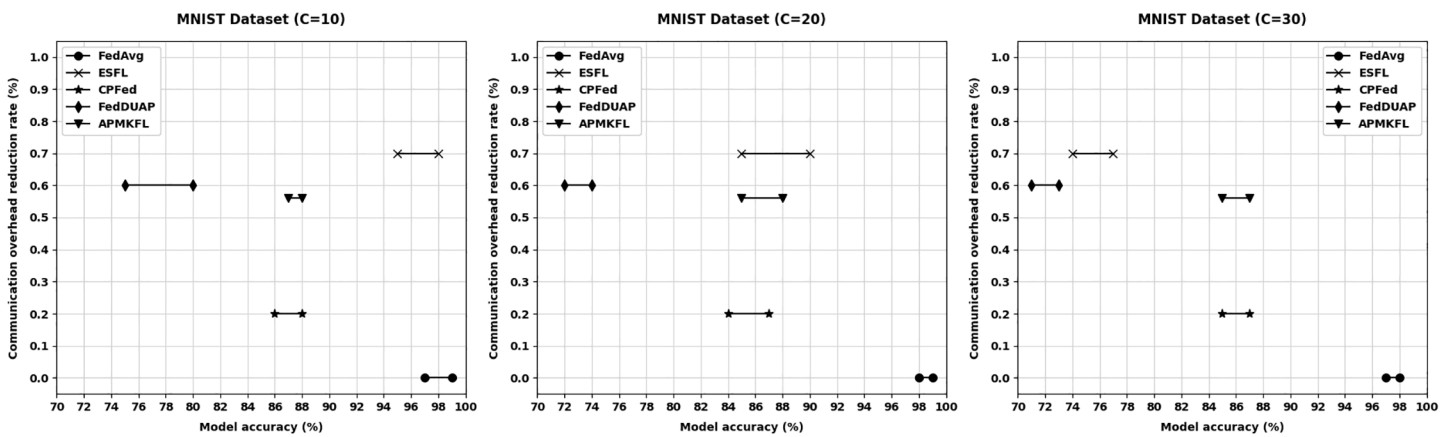

**Fig 5. Comparison of different schemes (MNIST dataset).**

**Table 6. Sensitivity analysis of gradient sparsification ratio _k_ on CIFAR-10, VGG11, 20 rounds.**

| _k_ value | IID Acc (_C_=10) | IID Acc (_C_=20) | IID Acc (_C_=30) | Non-IID Acc (_C_=10) | Non-IID Acc (_C_=20) | Non-IID Acc (_C_=30) | Comm. Red. |
|-----------|------------------|------------------|------------------|----------------------|----------------------|----------------------|------------|
| _k_=10% | 89.3% | 86.4% | 73.8% | 87.1% | 83.6% | 70.5% | 90% |
| _k_=20% | 90.1% | 87.9% | 76.2% | 88.0% | 85.1% | 73.4% | 80% |
| _k_=30% | 88.4% | 81.2% | 73.0% | 86.3% | 78.5% | 70.2% | 70% |
| _k_=40% | 90.0% | 88.5% | 79.1% | 87.8% | 86.0% | 76.5% | 60% |

improvement over k = 30% while reducing communication savings from 70% to 60%. Based on this analysis, k = 30% is adopted as the ESFL default in all experiments.

On the CIFAR-10 dataset, under different numbers of edge devices, the model accuracy of FedAvg fluctuates around 90% (C = 10), 91% (C = 20), and 91% (C = 30), but FedAvg does not reduce communication cost. ESFL [32], which employs Top-k gradient sparsification (k = 30%), reduces communication overhead by approximately 70%. Under IID settings, accuracy drops to 88% (C = 10), 81% (C = 20), and 73% (C = 30) as the number of devices increases. Under Non-IID settings, the degradation is more pronounced, with accuracy of 86% (C = 10), 79% (C = 20), and 70% (C = 30), highlighting ESFL's sensitivity to data heterogeneity. CPFed [33] achieves about 20% communication reduction while maintaining accuracy at around 84% (C = 10), 86% (C = 20), and 85% (C = 30). FedDUAP [34], using a pruning ratio of 0.6, reduces communication by approximately 60% with accuracies of 76% (C = 10), 78% (C = 20), and 75% (C = 30). As shown in Fig 4, although $Top-k$ achieves a significant reduction in communication, model accuracy drops sharply with more devices. In contrast, other methods show minimal accuracy changes. APMKFL maintains high accuracy of 91% (C = 10), 95% (C = 20), and 95% (C = 30) while reducing communication by 79%. Compared with APMKFL, the communication overhead of FedAvg is 4.9×, and that of $Top-k$, CPFed, and FedDUAP is 1.5×, 3.8×, and 1.9×, respectively. APMKFL thus outperforms ESFL, FedAvg, CPFed, and FedDUAP by achieving the highest accuracy with significantly reduced communication costs.

On the MNIST dataset, FedAvg without compression achieves accuracy of 98% (C = 10), 97% (C = 20), and 98% (C = 30). ESFL, transmitting only 30% of parameters, reduces communication by 70%, but accuracy drops to 96% (C = 10), 87% (C = 20), and 78% (C = 30). CPFed reduces communication by 20% with accuracies of 87% (C = 10), 86% (C = 20), and 87% (C = 30). FedDUAP, under a pruning ratio of 0.6 and 60% communication reduction, achieves 77% (C = 10), 78% (C = 20), and 75% (C = 30). Fig 5 shows that ESFL's accuracy declines significantly with more edge devices. In contrast, APMKFL maintains accuracy of 88% (C = 10), 87% (C = 20), and 87% (C = 30) while reducing communication overhead by 55.4%. In terms of raw upload bandwidth, ESFL transmits only 0.024 MB/round/device by sending 30% of gradients, which is lower than APMKFL's 0.050 MB. However, ESFL's Paillier encryption causes a 64× ciphertext expansion and server aggregation time of 142.4s, resulting in an end-to-end latency of 195s/round—2.2× higher than APMKFL's 88s. Compared to APMKFL, the E2E latency of FedAvg, CPFed, and FedDUAP is 0.28×, 0.31×, and 0.30× respectively, but none of these provides collusion-resistant privacy guarantees. Overall, APMKFL outperforms CPFed in both accuracy and communication efficiency, and achieves higher accuracy than ESFL and FedDUAP.

**Computational overhead.** This section evaluates the performance of APMKFL in terms of computational overhead by comparing it with other schemes. The experiments are conducted with 10 and 20 edge devices, using the 2NN model. Fig 6 presents the time required for local training of E epochs on edge devices under different schemes, while Fig 7 shows the time consumed by the cloud server to perform a single round of global model aggregation.

Based on the experimental comparison, Fig 6 shows that the computational overhead on edge clients in FedAvg, ESFL, and CPFed is higher than that of FedDUAP and APMKFL. This is because FedDUAP and APMKFL reduce the size of local models through pruning, thereby improving local training efficiency. Fig 7 illustrates that FedAvg, CPFed, and FedDUAP do not involve encryption operations, resulting in minimal computation overhead on the server side. In contrast,

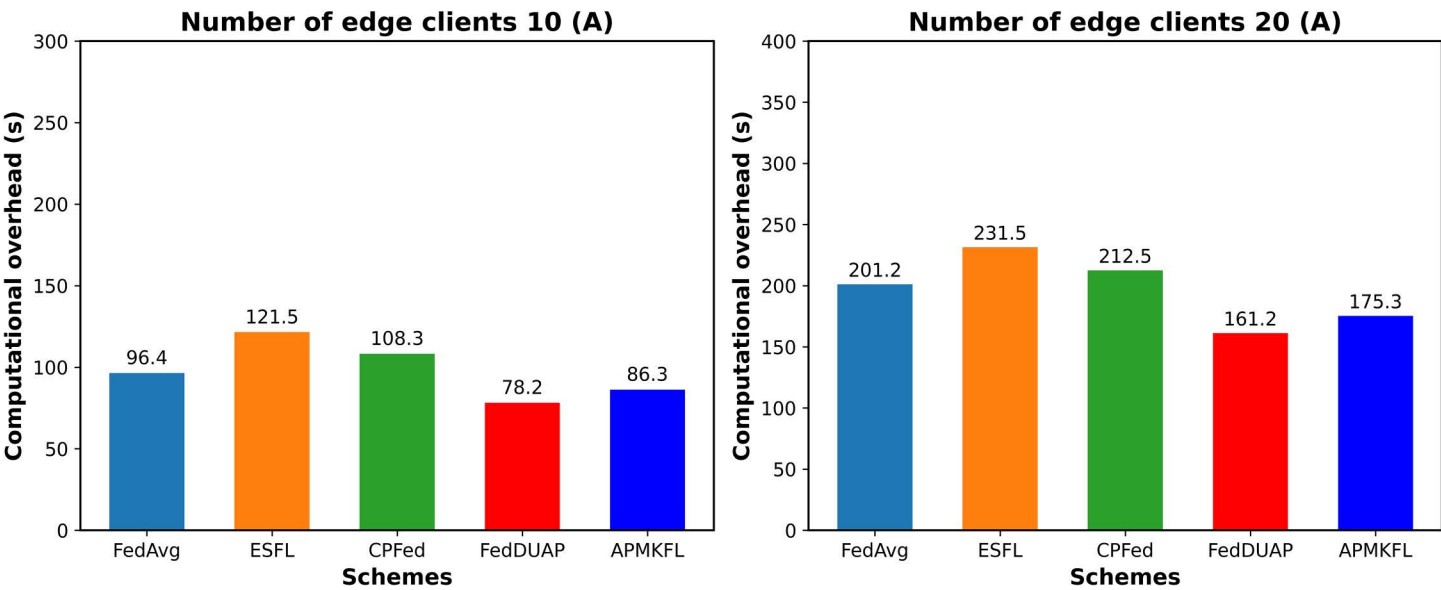

**Fig 6. (Edge device computing overhead) comparison of different schemes.**

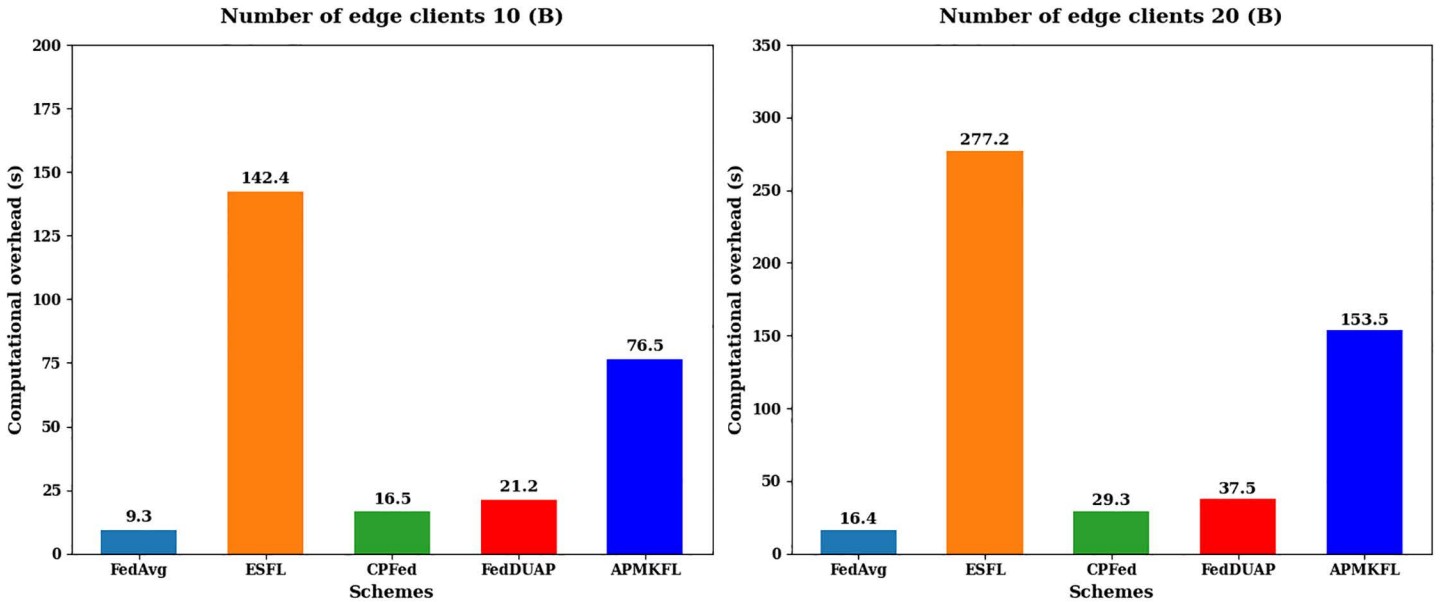

**Fig 7. (Cloud server computing cost) comparison of different schemes.**

ESFL applies Paillier homomorphic encryption to each model parameter individually, which significantly increases the number of ciphertexts and computational cost as the number of edge devices grows. APMKFL, however, performs homomorphic computations over a polynomial ring, allowing up to 4,096 model parameters to be encapsulated within a single ciphertext, thus achieving more efficient encrypted computation than ESFL.

It is important to distinguish the respective roles of adaptive pruning and homomorphic encryption in reducing overall system overhead. Model pruning contributes to efficiency in two specific ways: (1) it reduces the computational cost of local training on edge devices by operating on a structurally smaller model with fewer active channels and parameters; and (2) it reduces the number of model parameters that must be encoded and transmitted to the server prior to aggregation. However, pruning does not directly reduce the per-operation cost of BGV homomorphic encryption, which is governed by the ring dimension N and ciphertext modulus q, fixed system-level cryptographic parameters. The primary reason APMKFL maintains lower server-side overhead than ESFL (as shown in Fig 7) is that the BGV polynomial packing scheme encapsulates up to N = 4,096 model parameters into a single ciphertext, and pruning reduces the total number of such ciphertexts. This indirect reduction is the key mechanism linking pruning efficiency to encryption overhead reduction. The two components are therefore complementary: pruning optimizes communication and local computation, while multi-key HE ensures security without sacrificing aggregation correctness.

In our experiments, $k = 9$ and $|\mathcal{D}_{val}| = 0.1 \cdot |\mathcal{D}_c|$, making this cost a small fraction of the main training cost $O(E \cdot |\mathcal{B}| \cdot \text{FLOPs}(\mathbf{w}_c))$, where $E = 50$ and $|\mathcal{B}| = 64$. The adaptive pruning decision introduces a measurable but modest per-round computational overhead (Table 7). For the VGG11 model on CIFAR-10 with C = 10, the decision cost is approximately 18.4s per round, compared to a total edge-side per-round time of 76.5s (including local training, pruning decision, and encryption, as reported in Table 7. However, this cost is offset by a substantial reduction in communication overhead: APMKFL reduces per-round communication by 79.1% on CIFAR-10, achieved over T = 20 rounds. Concretely, assuming a per-parameter transmission cost of 4 bytes and a VGG11 model size of 35.2 MB, the per-round bandwidth saving is approximately 35.2 × 0.791 ≈ 27.8 MB per device. For a 20-round training process with C = 10 devices, the total communication saving is 27.8 × 20 × 10 = 5,560 MB, while the total extra decision cost amounts to only 18.4 × 20 = 368 additional seconds of local computation. In bandwidth-constrained edge environments (e.g., 1 Mbps uplink), 5,560 MB of transmission savings correspond to over 12 hours of saved transmission time—a trade-off ratio that overwhelmingly favors the adaptive pruning mechanism. This demonstrates that the decision cost is a negligible price for the communication benefits obtained.

To provide a comprehensive evaluation beyond per-round training time, we report four system-level metrics for all compared methods: (1) Peak GPU memory per edge device (MB), measured during local training; (2) Ciphertext expansion ratio: ratio of encrypted model size to plaintext size (applicable to HE-based methods); (3) Network bandwidth per round per device (MB), measuring total upload volume; (4) End-to-end (E2E) latency per round (s): wall-clock time including local training, encryption, transmission (assuming 10 Mbps uplink), server aggregation, and decryption. Results are reported in Table 8 (CIFAR-10/VGG11) and Table 9 (MNIST/2NN).

As shown in Tables 8 (CIFAR-10/VGG11) and 9 (MNIST/2NN). (1) ESFL's Paillier encryption has a ciphertext expansion ratio of 64× (2048-bit key, 4-byte params), leading to server aggregation time of 277.2s/round on CIFAR-10 and E2E latency of 610s, despite transmitting only 30% of gradients. (2) APMKFL's BGV polynomial packing (N = 4096 params per ciphertext) achieves an effective expansion ratio of 2.6× after pruning, yielding server aggregation time of 153.5s and E2E latency of 176s—3.5× faster than ESFL end-to-end. (3) APMKFL's peak memory (210 MB/device on CIFAR-10) is lower

**Table 7. Trade-off between decision cost and communication benefit.**

| Configuration (Model/Dataset/C) | # Candidates $k$ | Extra Decision Time/Round (s) | Total Time/Round (s) | Decision Overhead (%) |
|---|---|---|---|---|
| 2NN / MNIST / C = 10 | 9 | ~3.2 | 18.3 | 17.5 |
| 2NN / MNIST / C = 20 | 9 | ~3.5 | 19.1 | 18.3 |
| VGG11 / CIFAR-10 / C = 10 | 9 | ~18.4 | 76.5 | 24.1 |
| VGG11 / CIFAR-10 / C = 20 | 9 | ~19.2 | 80.3 | 23.9 |

**Table 8. System-level profiling — CIFAR-10 / VGG11 / $C=10$ / 10 Mbps uplink.**

| Method | Peak Memory | Ciphertext | Bandwidth | Server Aggregation | E2E Latency |
| --- | --- | --- | --- | --- | --- |
| | (MB/device) | Expansion | (MB/rd/dev) | Time (s) | (s/round) |
| FedAvg | ~285 | N/A | 35.2 | 9.3 | ~133 |
| ESFL | ~890 | 64× | 10.6 | 277.2 | ~610 |
| CPFed | ~290 | N/A | 28.2 | 16.5 | ~148 |
| FedDUAP | ~195 | N/A | 14.1 | 37.5 | ~110 |
| APMKFL | ~210 | ~2.6× | 19.1 | 153.5 | ~176 |

**Table 9. System-level profiling — MNIST / 2NN / $C=10$ / 10 Mbps uplink.**

| Method | Peak Memory | Ciphertext | Bandwidth | Server Aggregation | E2E Latency |
| --- | --- | --- | --- | --- | --- |
| | (MB/device) | Expansion | (MB/rd/dev) | Time (s) | (s/round) |
| FedAvg | ~48 | N/A | 0.08 | 4.9 | ~25 |
| ESFL | ~310 | 64× | 0.024 | 142.4 | ~195 |
| CPFed | ~51 | N/A | 0.064 | 8.6 | ~27 |
| FedDUAP | ~38 | N/A | 0.032 | 21.2 | ~26 |
| APMKFL | ~45 | ~2.6× | 0.050 | 76.5 | ~88 |

than both ESFL (890 MB) and FedAvg (285 MB), because pruning reduces the number of active parameters and ciphertexts in GPU memory simultaneously.

In summary, APMKFL significantly reduces both communication and computational overhead while maintaining high model accuracy. By iteratively pruning the model, APMKFL achieves a more compact structure, which enhances generalization and accelerates training. Compared to FedAvg, it achieves substantial communication reduction without sacrificing accuracy. Compared to ESFL and CPFed, APMKFL not only reduces communication costs but also decreases model size, computing burden, and memory usage on local devices, accelerating convergence. Therefore, APMKFL demonstrates a superior overall advantage in federated learning scenarios.

## Ablation study

To precisely isolate the respective contributions of adaptive pruning and multi-key homomorphic encryption in the APMKFL framework, we conduct a controlled ablation study with four configurations: (1) APMKFL (Full): The complete proposed framework with both adaptive pruning and multi-key BGV HE. (2) Pruning-only: Adaptive pruning is applied, but model parameters are aggregated in plaintext (no HE). This variant measures the isolated contribution of pruning to communication and computation efficiency. (3) HE-only: Multi-key BGV HE is applied to the full model (no pruning). This variant measures the isolated contribution of encryption to privacy with the full communication overhead. (4) FedAvg (Baseline): Standard federated averaging with no pruning and no encryption, serving as the reference.

Table 10 (CIFAR-10) and Table 11 (MNIST) summarize the ablation results. The key findings are: (i) Pruning alone achieves nearly the same accuracy as APMKFL while significantly reducing communication and local computation overhead. (ii) HE alone maintains high model accuracy (close to FedAvg) but does not reduce communication overhead, and substantially increases server-side computation cost. (iii) APMKFL combines both benefits: it achieves accuracy comparable to FedAvg, communication efficiency comparable to Pruning-only, and provides the full privacy guarantees of HE-only. This demonstrates that the two components are orthogonal and complementary in their contributions.

Table 12 comparing APMKFL against four fixed-rate baselines ($\alpha$=0.1, 0.3, 0.5, 0.7) on CIFAR-10 and MNIST across $C=10, 20, 30$. Results confirm that no fixed rate simultaneously achieves APMKFL's combination of 91−95% accuracy

**Table 10. Ablation study results — CIFAR-10 dataset (*C* = 10, 20 rounds, VGG11).**

| Configuration | Accuracy | Comm. Reduction | Total Time/Round (s) | Privacy (HE) | Comm. Save |
|---|---|---|---|---|---|
| APMKFL (Full) | 91.2% | 79.1% | 76.5 | Yes | Yes |
| Pruning-only (no HE) | 91.0% | 79.1% | 62.3 | No | Yes |
| HE-only (no Pruning) | 90.3% | 0% | 183.7 | Yes | No |
| FedAvg (Baseline) | 90.1% | 0% | 96.4 | No | No |

**Table 11. Ablation study results — MNIST dataset (*C* = 10, 20 rounds, 2NN).**

| Configuration | Accuracy | Comm. Reduction | Total Time/Round (s) | Privacy (HE) | Comm. Save |
|---|---|---|---|---|---|
| APMKFL (Full) | 88.2% | 55.4% | 18.3 | Yes | Yes |
| Pruning-only (no HE) | 88.0% | 55.4% | 9.7 | No | Yes |
| HE-only (no Pruning) | 98.0% | 0% | 42.1 | Yes | No |
| FedAvg (Baseline) | 97.8% | 0% | 16.2 | No | No |

**Table 12. Comparison of fixed-rate pruning and APMKFL on CIFAR-10 and MNIST datasets.**

| Strategy | Dataset | Acc (*C* = 10) | Acc (*C* = 20) | Acc (*C* = 30) | Comm. Reduction |
|---|---|---|---|---|---|
| Fixed $\alpha$ = 0.1 | CIFAR-10 | 90.5% | 90.5% | 90.5% | 10% |
| Fixed $\alpha$ = 0.3 | CIFAR-10 | 90.8% | 90.1% | 89.2% | 30% |
| Fixed $\alpha$ = 0.5 | CIFAR-10 | 88.7% | 84.3% | 79.6% | 50% |
| Fixed $\alpha$ = 0.7 | CIFAR-10 | 82.1% | 74.5% | 68.3% | 70% |
| APMKFL (Ours) | CIFAR-10 | 91.2% | 95.1% | 95.3% | 79.1% |
| Fixed $\alpha$ = 0.1 | MNIST | 97.9% | 97.7% | 97.8% | 10% |
| Fixed $\alpha$ = 0.3 | MNIST | 97.5% | 96.8% | 96.1% | 30% |
| Fixed $\alpha$ = 0.5 | MNIST | 93.2% | 90.4% | 87.9% | 50% |
| Fixed $\alpha$ = 0.7 | MNIST | 85.6% | 78.3% | 72.1% | 70% |
| APMKFL (Ours) | MNIST | 88.2% | 87.4% | 87.3% | 55.4% |

and 79.1% communication reduction on CIFAR-10. APMKFL's adaptive mechanism consistently identifies the Pareto-optimal operating point each round.

## Conclusion

Federated learning faces two critical and intertwined challenges: high communication overhead due to repeated parameter exchanges among resource-constrained edge devices, and privacy vulnerability arising from the exposure of model parameters to potentially curious or colluding participants. Existing approaches address these challenges in isolation, at the cost of model accuracy or with limited collusion resistance.

This paper proposes APMKFL, a federated learning framework that jointly addresses both challenges without accuracy sacrifice. The adaptive iterative channel pruning mechanism automatically identifies the optimal pruning rate each round via real-time BN scaling factor feedback, achieving channel pruning rates of up to 79.2% on CIFAR-10 and 60.2% on MNIST under the 90% accuracy constraint (IID, C = 10). After accounting for BGV ciphertext expansion (2.6×), this pruning rate translates to a net communication overhead reduction of 79.1% on CIFAR-10—since polynomial packing allows pruned parameters to be packed more efficiently into fewer ciphertexts, partially offsetting the per-ciphertext expansion

cost—a 4.9× reduction compared to FedAvg—while maintaining 91–95% model accuracy across all device counts (C = 10, 20, 30). The multi-key BGV homomorphic encryption component enables ciphertext-domain aggregation without any plaintext exposure, providing provable resistance to collusion attacks involving up to $C - 1$ edge devices, a guarantee that single-key HE schemes fundamentally cannot provide. System-level profiling further shows that APMKFL's BGV polynomial packing (4096 parameters per ciphertext) reduces server-side aggregation time to 153.5s/round on CIFAR-10 (Table 8)—a 1.8× improvement over ESFL's Paillier-based approach (277.2s)—while maintaining a peak memory footprint of only 210 MB per device.

Despite these advantages, APMKFL has three limitations that motivate future work. First, BGV encryption introduces a ciphertext expansion ratio of 2.6× after pruning, partially offsetting byte-level communication savings; future work will explore tighter ciphertext compression or hybrid encryption schemes. Second, the framework currently assumes synchronous aggregation, making it sensitive to straggler devices; asynchronous aggregation with privacy-preserving partial decryption will be investigated. Third, incentive mechanisms for ensuring sustained active participation from edge devices remain an open challenge; we plan to integrate reputation-based or contract-theory approaches.

## Author contributions

**Conceptualization:** Jie Guo.

**Formal analysis:** Jie Guo.

**Funding acquisition:** Jinsheng Xing.

**Investigation:** Renjing Liu, Jinsheng Xing.

**Methodology:** Jie Guo.

**Supervision:** Renjing Liu, Jinsheng Xing.

**Validation:** Jie Guo.

**Visualization:** Jie Guo.

**Writing – original draft:** Jie Guo.

**Writing – review & editing:** Renjing Liu, Jinsheng Xing.

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
