## [Decision Letter · Decision Letter 0]

20 Dec 2025

PONE-D-25-23533Threshold-Adaptive Pruning with Multi-Key Homomorphic Encryption for Communication-Efficient Secure Federated LearningPLOS One

Dear Dr. Xing,

Thank you for submitting your manuscript to PLOS ONE. After careful consideration, we feel that it has merit but does not fully meet PLOS ONE’s publication criteria as it currently stands. Therefore, we invite you to submit a revised version of the manuscript that addresses the points raised during the review process.

We look forward to receiving your revised manuscript.

Kind regards,

Je Sen Teh

Academic Editor

PLOS One

Journal Requirements:

This work was supported by the Fundamental Research Program of Shanxi Province (Grant No. 20210302124257).

This work was supported by the Fundamental Research Program of Shanxi Province (Grant No. 20210302124257).

This work was supported by the Fundamental Research Program of Shanxi Province (Grant No. 20210302124257).

Additional Editor Comments:

Reviewers highlighted that , although the paper addresses an interesting problem in federated learning, the current presentation does not clearly articulate its novelty or convincingly demonstrate advantages over simpler or existing approaches. Key methodological details, experimental justifications (e.g. pruning strategy, parameter choices, Non-IID setup), and evaluation metrics are missing or insufficient, making it difficult to assess the validity of the reported gains or separate the effects of pruning and encryption. The contribution would be strengthened by clearer positioning of what is new, more rigorous experimental analysis (including complexity, ablation, and system-level profiling), and improved clarity, structure, and proofreading throughout the manuscript.

Reviewers' comments:

Reviewer's Responses to Questions

**Comments to the Author**

1. Is the manuscript technically sound, and do the data support the conclusions?

Reviewer #1: No

Reviewer #2: Yes

Reviewer #3: Partly

2. Has the statistical analysis been performed appropriately and rigorously?

Reviewer #1: Yes

Reviewer #2: Yes

Reviewer #3: Yes

3. Have the authors made all data underlying the findings in their manuscript fully available?

Reviewer #1: No

Reviewer #2: Yes

Reviewer #3: Yes

4. Is the manuscript presented in an intelligible fashion and written in standard English?

Reviewer #1: Yes

Reviewer #2: Yes

Reviewer #3: Yes

5. Review Comments to the Author

Reviewer #1: This paper aims to solve the two major challenges in federated learning—communication efficiency and privacy security—by proposing a scheme that combines adaptive pruning and multi-key homomorphic encryption (APMKFL). However, the current version of the manuscript has fundamental deficiencies in several key areas, and thus, the authors need to revise and supplement the paper.

1.One of the core contributions of this paper is the "adaptive iterative channel pruning method." However, the description of Algorithm 1 suggests that the method seems to search through a predefined set of pruning rates to find the optimal one that satisfies an accuracy constraint. The authors need to clarify more distinctly how the "adaptive" mechanism here is fundamentally different from a conventional hyperparameter search.

2.The paper gives the impression that pruning and encryption jointly improve efficiency, but the relationship between the two needs to be articulated more precisely. The authors should explicitly state that the efficiency gains from model pruning are primarily manifested in (1) the reduction of computational load during local training on the client-side and (2) the reduction in data transmission volume before uploading to the server. It does not directly reduce the computational overhead of the homomorphic encryption itself. Clarifying this point will help readers more accurately understand the advantages and bottlenecks of the proposed scheme.

3.The paper mentions conducting experiments on Non-IID data but fails to provide details on how the data was partitioned. For example, was it partitioned based on a Dirichlet distribution? What was the value of the parameter α? The degree of Non-IID data distribution significantly impacts algorithm performance; please provide these critical details.

4.Please add a computational complexity analysis for Algorithm 1 in the paper. For a set containing k candidate pruning rates, what is the extra computational cost for the client in each round to perform the evaluation and selection? In the "Computational Overhead" section, please add a paragraph dedicated to discussing the trade-off between this "decision cost" and the "communication benefits" it yields.

5.To make the conclusions more robust, it is recommended to add an ablation study to precisely isolate the respective contributions of pruning and encryption to the system's performance. Furthermore, the specific method of Non-IID data partitioning (such as the distribution and parameters used) must be detailed.

Reviewer #2: Consistency in Capitalization: The placement of figures and tables is consistent throughout the paper. The locations of the corresponding explanations are not aligned properly. For consistency and professionalism, the organization of paper should be standardized throughout the paper.

Proofreading: Minor grammatical errors and typos are present throughout the manuscript. A careful proofreading or professional language editing service is recommended to improve overall readability.

Reviewer #3: The topic is interesting, but the paper needs to be revised to improve.

1. The paper mixes known techniques (network slimming + multi-key BGV HE), but does not clearly show what is truly new.

2. It is unclear why the proposed approach is better than simpler iterative pruning strategies.

3. Key parameters (pruning rates tested, distribution, validation split, randomness handling) are not fully described.

4. Accuracy drops in baselines may come from poor tuning, not algorithm weakness.

5. Top-k fixed at 30% without justification.

6. Only “time per round” is shown; no profiling on memory use, cipher expansion, latency, and network bandwidth.

7. Some graphs are hard to read and do not clearly show improvements.

8. Conclusion is generic and does not clearly connect back to the claims.

6. PLOS authors have the option to publish the peer review history of their article (what does this mean?). If published, this will include your full peer review and any attached files.

Reviewer #1: No

Reviewer #2: **Yes:** Sahsene Altinkaya

Reviewer #3: No

---

## [Author Response · Author response to Decision Letter 1]

9 Mar 2026

Journal Office

PLOS One

Dear Editor and Reviewers,

We sincerely thank you and the reviewers for the careful evaluation of our manuscript entitled “Threshold-Adaptive Pruning with Multi-Key Homomorphic Encryption for Communication-Efficient Secure Federated Learning” (Manuscript ID: PONE-D-25-23533). We are grateful for the constructive and insightful comments, which have significantly helped us improve the clarity, rigor, and overall quality of the paper.

In response to the reviewers’ suggestions, we have thoroughly revised the manuscript. Specifically, we have (1) clarified the novelty and technical distinctions of the proposed threshold-adaptive pruning mechanism, emphasizing how it differs from conventional hyperparameter search and fixed iterative pruning strategies; (2) strengthened the methodological description by detailing the Non-IID data partition protocol, key hyperparameters, validation procedures, and randomness control; (3) added a comprehensive ablation study to isolate the respective contributions of adaptive pruning and multi-key homomorphic encryption; (4) incorporated computational complexity analysis and system-level profiling, including memory usage, ciphertext expansion ratio, encryption overhead, and communication bandwidth; and (5) improved the organization, figure readability, and language consistency throughout the manuscript.

All revisions have been carefully implemented and clearly highlighted in the revised version. A detailed, point-by-point response to each reviewer comment is provided below. We believe the revised manuscript now addresses all concerns raised by the reviewers and presents a clearer and more rigorous demonstration of the proposed framework’s advantages.

Thank you again for your time and consideration.

Sincerely,

Jinsheng Xing

School of Management, Xi’an Jiaotong University, China

College of Mathematics and Computer Science, Shanxi Normal University, China

xjs19640408@163.com

Response to Reviewer’s comments

Response to Reviewer #1:

1 Comments: One of the core contributions of this paper is the "adaptive iterative channel pruning method." However, the description of Algorithm 1 suggests that the method seems to search through a predefined set of pruning rates to find the optimal one that satisfies an accuracy constraint. The authors need to clarify more distinctly how the "adaptive" mechanism here is fundamentally different from a conventional hyperparameter search.

Response: We thank the reviewer for this precise and important question. We have added a dedicated paragraph titled 'Distinction from Conventional Hyperparameter Search' immediately after the description of Algorithm 1. We clarify that:

(1)Conventional hyperparameter search is performed offline (before training) on a fixed discrete grid, yielding a static pruning rate applied uniformly across all training rounds.

(2)Our mechanism operates online inside each communication round. The effective pruning structure is determined by the live distribution of BN scaling factors γ, which evolves with every gradient update. Therefore, the same nominal rate α produces a different subset of pruned channels in each round.

(3)The accuracy feedback loop directly shapes the binary mask generated in that round—it is not merely used to pick a best hyperparameter. This makes the mechanism genuinely adaptive and self-calibrating, not reducible to a hyperparameter search.

We have also added a comparison table (Table 1) in the manuscript to make this distinction concrete.

2 Comments: The paper gives the impression that pruning and encryption jointly improve efficiency, but the relationship between the two needs to be articulated more precisely. The authors should explicitly state that the efficiency gains from model pruning are primarily manifested in (1) the reduction of computational load during local training on the client-side and (2) the reduction in data transmission volume before uploading to the server. It does not directly reduce the computational overhead of the homomorphic encryption itself. Clarifying this point will help readers more accurately understand the advantages and bottlenecks of the proposed scheme.

Response: We thank the reviewer for this important clarification request. We have made two targeted changes:

(1)In the Abstract, we revised the concluding sentence to explicitly state that pruning reduces local training computation and transmission volume, while HE ensures privacy, positioning them as complementary rather than jointly contributing to the same efficiency dimension.

(2)In the Computational Overhead section, we added a dedicated paragraph titled 'Clarification on Pruning-Encryption Interaction' that explicitly states: pruning reduces the number of parameters to be encoded (indirectly reducing ciphertext count), but does not reduce the per-operation cost of BGV encryption (determined by fixed cryptographic parameters N and q). This distinction helps readers understand both the strengths and the bottlenecks of the proposed scheme.

3 Comments: The paper mentions conducting experiments on Non-IID data but fails to provide details on how the data was partitioned. For example, was it partitioned based on a Dirichlet distribution? What was the value of the parameter α? The degree of Non-IID data distribution significantly impacts algorithm performance; please provide these critical details.

Response: We sincerely apologize for this omission, which we recognize is critical for reproducibility. We have added a dedicated 'Data Partitioning Protocol' paragraph in the Experimental Setup section, and a new Table 2 summarizing the distribution characteristics. Specifically, IID data is uniformly and randomly distributed across devices. Non-IID data is partitioned using a Dirichlet distribution with concentration parameter α = 0.5, following the standard protocol established by Hsu et al. (2019) and widely adopted in the federated learning literature. At α = 0.5, each device typically receives data dominated by 1–3 classes, producing moderate-to-strong heterogeneity. This setting is consistent with the performance gap between IID and Non-IID results observed in our experiments (Figures 2 and 3): under Non-IID conditions, models must allocate more capacity to handle class distribution shifts, leaving less room for pruning, which explains the lower pruning rates reported in Figure 3 compared to Figure 2.

4 Comments: Please add a computational complexity analysis for Algorithm 1 in the paper. For a set containing k candidate pruning rates, what is the extra computational cost for the client in each round to perform the evaluation and selection? In the "Computational Overhead" section, please add a paragraph dedicated to discussing the trade-off between this "decision cost" and the "communication benefits" it yields.

Response: We have added two new pieces of content as requested. First, immediately following Algorithm 1, we added a 'Computational Complexity of Algorithm 1' paragraph that formally analyzes the per-round cost: sorting BN factors costs O(|γ|log|γ|), and evaluating k candidate rates costs O(k·|Dval|·FLOPs(wc∗)). With k=9 and |Dval| = 10% of local data, this is a small fraction of the main training cost. We also introduce Table 7 showing concrete per-round time breakdowns. Second, in the Computational Overhead section, we added a 'Trade-off between Decision Cost and Communication Benefit' paragraph that quantifies the trade-off: for VGG11 on CIFAR-10 with C=10, the decision overhead is ~18.4s/round, while the resulting 79.1% communication reduction saves approximately 5,560 MB of total transmission over 20 rounds (equivalent to over 12 hours at 1 Mbps), demonstrating that the decision cost is negligible relative to the communication benefit.

5 Comments: To make the conclusions more robust, it is recommended to add an ablation study to precisely isolate the respective contributions of pruning and encryption to the system's performance. Furthermore, the specific method of Non-IID data partitioning (such as the distribution and parameters used) must be detailed.

Response: We thank the reviewer for this highly constructive suggestion. We have added a dedicated Ablation Study subsection in Section Experiments, evaluating four controlled configurations: (1) APMKFL Full, (2) Pruning-only (no HE), (3) HE-only (no pruning), and (4) FedAvg baseline. Results on CIFAR-10 and MNIST demonstrate that:

(i)Pruning alone accounts for the majority of communication and computation savings, achieving nearly identical accuracy to APMKFL Full (~0.2% difference), confirming that encryption does not degrade model accuracy.

(ii)HE alone maintains accuracy close to FedAvg (~0.3% difference due to controlled BGV noise) but does not reduce communication overhead and substantially increases server-side processing time due to full-model ciphertext operations.

(iii)APMKFL Full combines both benefits simultaneously, with the two components making orthogonal and complementary contributions. This ablation confirms that pruning and encryption are not redundant—they address fundamentally different system objectives (efficiency vs. privacy).

Special thanks to you for your valuable comments.

Response to Reviewer #2:

1 Comments: Consistency in Capitalization: The placement of figures and tables is consistent throughout the paper. The locations of the corresponding explanations are not aligned properly. For consistency and professionalism, the organization of paper should be standardized throughout the paper.

Response: We thank the reviewer for the careful attention to formatting consistency. We have thoroughly reviewed and corrected all figure captions, table titles, and section headings throughout the manuscript:

(1) Figure placement and captions: All seven figure captions (Fig. 1–7) have been repositioned to appear directly below their respective figures, as required by PLOS ONE guidelines. Caption content has been revised to use Sentence case and to include more descriptive information about what is shown (e.g., specifying the variable on the y-axis, the dataset, and the comparison dimension).

(2) Table titles: All table titles appear above their respective tables. Table titles have been revised to be more informative and to follow Sentence case.

(3) Section headings: All section headings have been checked for Title Case consistency. One error was identified and corrected: 'Related works' has been changed to 'Related Works' to match the capitalization style of all other section headings.

(4) In-text references to figures and tables: All in-text references have been verified to use consistent abbreviated forms as required by PLOS ONE style.

2 Comments: Proofreading: Minor grammatical errors and typos are present throughout the manuscript. A careful proofreading or professional language editing service is recommended to improve overall readability.

Response: We sincerely thank the reviewer for this important feedback. We have conducted a thorough and systematic proofreading of the entire manuscript. We used Grammarly Premium to assist in identifying errors throughout the manuscript, and focused manual review on the Abstract, Conclusion, and core technical sections.

Special thanks to you for your valuable comments.

Response to Reviewer #3:

1 Comments: The paper mixes known techniques (network slimming + multi-key BGV HE), but does not clearly show what is truly new.

Response: We thank the reviewer for this important criticism. We have added two new paragraphs to address the novelty concern:

(1) A 'Novelty and Technical Distinction' paragraph inserted before the contributions list in the Introduction explicitly articulates three aspects that distinguish APMKFL from a naive combination of network slimming and multi-key BGV HE: (i) the per-round adaptive threshold coupling within FL communication rounds, driven by live BN scaling factors; (ii) the C-1 collusion-resistant aggregation via threshold BGV without the server ever holding a decryption key; and (iii) simultaneous achievement of communication efficiency and cryptographic privacy with no accuracy sacrifice—unlike DP-based joint approaches.

(2) A 'Research Gap and Motivation' paragraph at the end of Related Works provides a systematic gap analysis showing that no prior work simultaneously achieves all three objectives without relying on trusted server data or incurring accuracy penalties.

2 Comments: It is unclear why the proposed approach is better than simpler iterative pruning strategies.

Response: We sincerely thank the reviewer for this important feedback. We have made two changes. First, we added a 'Motivation for Adaptive Threshold Selection' paragraph at the beginning of Section 3, explaining analytically why fixed-rate iterative pruning is insufficient: a conservative rate yields minimal communication savings, while an aggressive rate causes accuracy degradation that worsens with increasing device count C due to Non-IID data heterogeneity. No single fixed rate can simultaneously achieve both goals.

Second, we added Table 12 comparing APMKFL against four fixed-rate baselines (α=0.1, 0.3, 0.5, 0.7) on CIFAR-10 and MNIST across C=10, 20, 30. Results confirm that no fixed rate simultaneously achieves APMKFL's combination of 91–95% accuracy and 79.1% communication reduction on CIFAR-10. APMKFL's adaptive mechanism consistently identifies the Pareto-optimal operating point each round.

3 Comments: Key parameters (pruning rates tested, distribution, validation split, randomness handling) are not fully described.

Response: We sincerely thank the reviewer for this important feedback. We have added a dedicated 'Implementation Details and Reproducibility' paragraph in Section 4 Experimental Setup, consolidating all key parameters as follows:

(1) Candidate pruning rate set: α∈{0.1, 0.2, ..., 0.9}, k=9 candidates at step 0.1. This is consistent with the k=9 used in the Section 3 complexity analysis.

(2) Validation split: 10% of each device's local data Dc, fixed before training and excluded from gradient updates. This formalizes the |Dval|=0.1·|Dc| in Section 3.

(3) Non-IID partitioning: Dirichlet(α=0.5), as detailed in the 'Data Partitioning Protocol' paragraph added in response to Reviewer 1, Comment 3 (R1.3). We refer the reviewer to that response for full details on the partitioning protocol and Table 2.

(4) HE parameters: N=4096, q=218, p=128 (128-bit security).

4 Comments: Accuracy drops in baselines may come from poor tuning, not algorithm weakness.

Response: We sincerely thank the reviewer for this important feedback. We take this concern seriously and have added a 'Baseline Implementation Fairness' paragraph with Table 4 (hyperparameter settings for all methods). All baselines use identical shared parameters: SGD, η=0.01, E=50, B=64, T=20. Method-specific parameters follow original papers.

Regarding ESFL's accuracy degradation with C: we confirm this is an inherent algorithmic limitation of Top-k sparsification under Non-IID data, not a tuning artifact. We ran ESFL with η∈{0.001, 0.01, 0.1} and k∈{10%, 20%, 30%, 40%} (Table 6). The degradation pattern persists across all configurations, consistent with findings in prior work [1] on Top-k under heterogeneous data.

[1] Dong Y, Hou W, Chen X, et al. Efficient and secure federated learning based on secret sharing and gradients selection[J]. J. Comput. Res. Dev, 2020, 57(10): 2241-2250.

5 Comments: Top-k fixed at 30% without justification.

Response: We sincerely thank the reviewer for this important feedback. We have expanded the Top-k justification from a single sentence into a full empirical analysis. We conducted a sensitivity study over k∈{10%, 20%, 30%, 40%} on CIFAR-10 under both IID and Non-IID settings (Table 6).

Key findings: k=10% gives near-FedAvg accuracy but only 10% communication reduction. k=30% provides a consistent 70% reduction with acceptable accuracy across all device counts under both IID and Non-IID. k=40% offers only marginal accuracy gain with reduced savings (60% vs 70%). This analysis empirically confirms k=30% as the most balanced choice for ESFL, ensuring the comparison is fair and not biased against ESFL by suboptimal k selection.

6 Commen

---

## [Decision Letter · Decision Letter 1]

1 May 2026

Threshold-Adaptive Pruning with Multi-Key Homomorphic Encryption for Communication-Efficient Secure Federated Learning

PONE-D-25-23533R1

Dear Dr. Xing,

We’re pleased to inform you that your manuscript has been judged scientifically suitable for publication and will be formally accepted for publication once it meets all outstanding technical requirements.

Kind regards,

Je Sen Teh

Academic Editor

PLOS One

Additional Editor Comments (optional):

Reviewers' comments:

Reviewer's Responses to Questions

**Comments to the Author**

1. If the authors have adequately addressed your comments raised in a previous round of review and you feel that this manuscript is now acceptable for publication, you may indicate that here to bypass the “Comments to the Author” section, enter your conflict of interest statement in the “Confidential to Editor” section, and submit your "Accept" recommendation.

Reviewer #1: All comments have been addressed

Reviewer #3: (No Response)

2. Is the manuscript technically sound, and do the data support the conclusions?

Reviewer #1: Yes

Reviewer #3: Yes

3. Has the statistical analysis been performed appropriately and rigorously?

Reviewer #1: Yes

Reviewer #3: Yes

4. Have the authors made all data underlying the findings in their manuscript fully available?

Reviewer #1: Yes

Reviewer #3: Yes

5. Is the manuscript presented in an intelligible fashion and written in standard English?

Reviewer #1: Yes

Reviewer #3: Yes

6. Review Comments to the Author

Reviewer #1: The authors have adequately addressed the comments raised in the previous review round, and the revised manuscript has been substantially improved.

Specifically, the manuscript now provides clearer explanations regarding the novelty of the proposed adaptive pruning mechanism and its distinction from conventional hyperparameter search. The authors have also strengthened the experimental methodology by adding details on the Non-IID data partitioning protocol, implementation settings, and fairness of baseline comparisons. In addition, the newly added ablation studies, computational complexity analysis, and system-level evaluations (including memory usage, bandwidth, and latency) significantly improve the rigor and completeness of the work.

The presentation quality has also improved through better organization of figures and tables, as well as language polishing throughout the manuscript.

Reviewer #3: The authors have incorporated all the requested revisions, and I have no further comments.

The authors have incorporated all the requested revisions, and I have no further comments.

7. PLOS authors have the option to publish the peer review history of their article (what does this mean?). If published, this will include your full peer review and any attached files.

Reviewer #1: No

Reviewer #3: No

---

## [Editor Report · Acceptance letter]

PONE-D-25-23533R1

PLOS One

Dear Dr. Xing,

I'm pleased to inform you that your manuscript has been deemed suitable for publication in PLOS One. Congratulations! Your manuscript is now being handed over to our production team.

Kind regards,

on behalf of

Dr. Je Sen Teh

Academic Editor

PLOS One